# GDI-Bench: A Benchmark for General Document Intelligence with Vision and Reasoning Decoupling

## Abstract

The rapid advancement of multimodal large language models (MLLMs) has profoundly impacted the document domain, creating a wide array of application scenarios. This progress highlights the need for a comprehensive benchmark to evaluate these models' capabilities across various document-specific tasks. However, existing benchmarks often fail to locate specific model weaknesses or guide systematic improvements. To bridge this gap, we introduce a General Document Intelligence Benchmark (GDI-Bench), featuring 2.3k images across 9 key scenarios and 19 document-specific tasks. By decoupling visual complexity and reasoning complexity, the GDI-Bench structures graded tasks that allow performance assessment by difficulty, aiding in model weakness identification and optimization guidance. We evaluate various open-source and closed-source models on GDI-Bench, conducting decoupled analyses in the visual and reasoning domains, revealing their strengths and weaknesses. To address the diverse tasks and domains in the GDI-Bench, we propose a GDI-Model that mitigates catastrophic forgetting during the supervised fine-tuning (SFT) process through an intelligence-preserving training strategy, thereby reinforcing the inherent weaknesses of the base model. Our model achieves state-of-the-art performance on previous benchmarks and the GDI-Bench. Both our benchmark and models are open-sourced on `https://huggingface.co/GDIBench`.

## 1 Introduction

Rapid progress of large language models (LLMs) Liu et al. (2024a); Guo et al. (2025) has placed multimodal large language models (MLLMs) Bai et al. (2023); OpenAI (2023); Team et al. (2023) as a foundation of artificial intelligence, advancing document intelligence to a general stage. Cross-domain and multi-scale document understanding and extraction challenges are increasingly critical in real-world applications. With the rise of MLLMs, a series of more complex benchmarks have emerged to provide comprehensive frameworks for document understanding tasks Fu et al. (2024a); Wu et al. (2024); Guan et al. (2024); Li et al. (2023a; 2024); Zhao et al. (2024); Feng et al. (2024). However, given that document understanding involves multiple modalities, errors in MLLM outputs may arise from inaccurate visual recognition, limited language organization, or both. Consequently, a decoupled evaluation of MLLMs' document processing abilities is essential.

To address these challenges, we propose the General Document Intelligence Benchmark (GDI-Bench), illustrated in Fig. 1, which aims to perform a decoupled evaluation of model performance on document tasks, thereby contributing to the identification of the model's weaknesses. The GDI-Bench introduces three key improvements over existing benchmarks: (1) developing a cross-domain, multi-task benchmarks to ensure task diversity and fine-grained difficulty levels; (2) introducing complexity decoupling, dividing multimodal document understanding into visual complexity and reasoning complexity, and for the first time, establishing a difficulty grading mechanism; (3) supporting the evaluation of MLLMs, OCR+LLM-level systems, and document parsing tools, offering comprehensive guidance for practical application solutions.

We evaluate 2 open-source and 4 closed-source large-scale models using the GDI-Bench and find their performance to be suboptimal, as shown in Fig. 2. For instance, while the InternVL3-8B

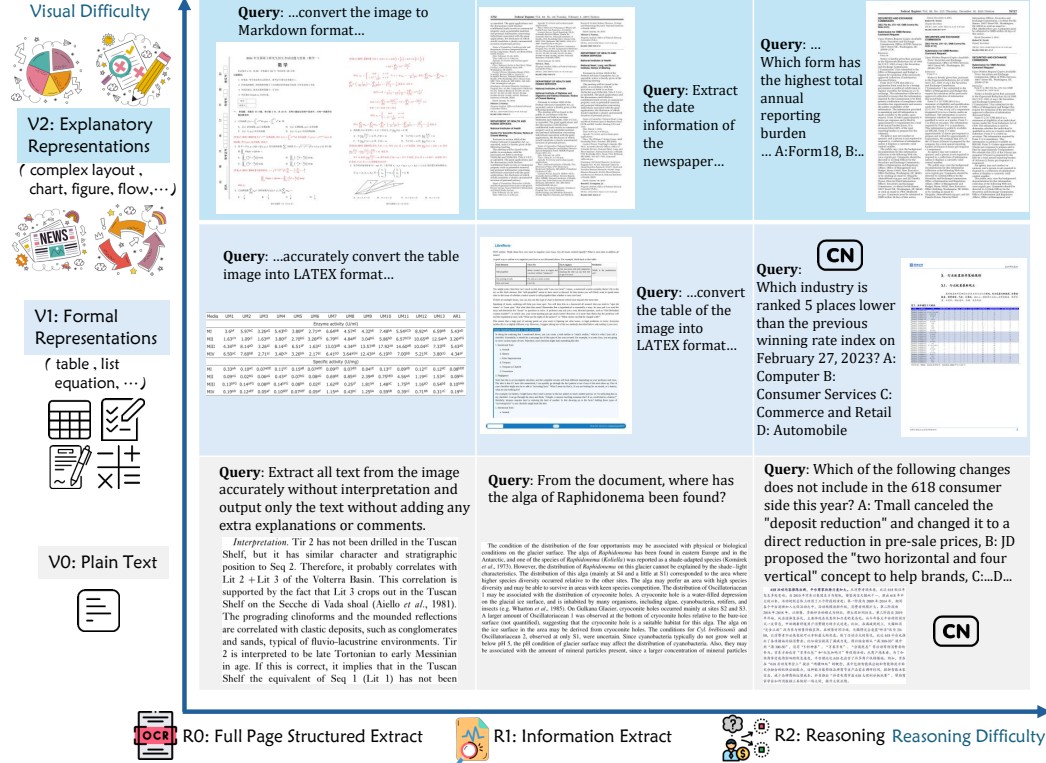

Figure 1: Overview of GDI-Bench. The benchmark decouples document understanding complexity into visual complexity (V0-V2) and reasoning complexity (R0-R2) dimensions, creating a comprehensive evaluation framework for assessing MLLMs' capabilities across various document types and reasoning tasks. Queries marked with a "CN" tag originate in Chinese and have been translated into English using Google Translate.

model performs well in the R0 domain but shows significant performance degradation in the R1 and R2 domains To address the model's weaknesses and further validate GDI-Bench's capacity for weakness localization, we constructed supervised fine-tuning (SFT) on the InternVL3-8B model to explore whether data-driven approaches could improve its performance. However, we observe that supervised fine-tuning tends to cause the issue of catastrophic forgetting Kirkpatrick et al. (2017). To address this problem, we propose the Layer-wise Adaptive Freezing-Tuning (LW-AFT) method, which alleviates the impact of catastrophic forgetting and enhances the model's cross-domain and cross-task capabilities. Specifically, during the SFT process, LW-AFT freezes most of the parameters, with only a small subset of domain-sensitive parameters participating in gradient updates.

Our contributions are summarized as follows.

- We propose the GDI-Bench, a benchmark covering a broad range of document-related tasks. By decoupling complexity and grading difficulty, it helps the model identify its weaknesses and guides subsequent optimization.

- We propose Layer-wise Adaptive Freeze-Tuning, a training method that effectively alleviates catastrophic forgetting in document task SFT by parameter-freezing, improving performance on specific tasks while maintaining generalization capabilities.

- We propose a GDI-Model that achieves state-of-the-art (SOTA) performance on multiple document domain benchmarks as well as the GDI-Bench, demonstrating high generalization capabilities suitable for real-world applications.

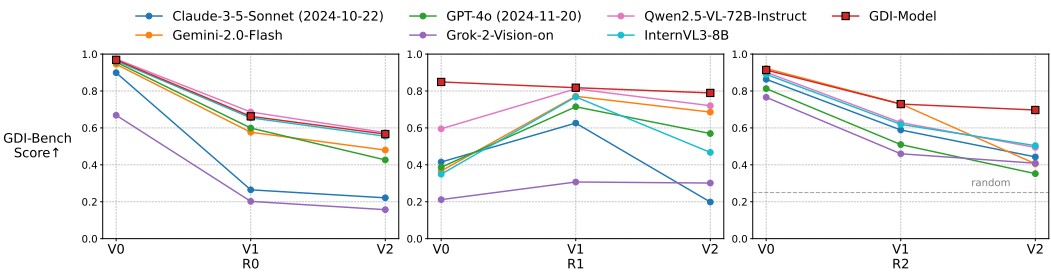

Figure 2: Performance of various open-source and closed-source models on GDI-Bench at different levels of reasoning complexity. The GDI-Model is fine-tuned based on the InternVL3-8B model. The gray line denotes the expected performance of random guessing in R2 tasks.

## 2 RELATED WORKS

**Document Benchmark**  Early benchmarks in document understanding primarily focused on single-domain tasks. For example, DocVQAQi et al. (2022) tackled industrial document QA, VisualMRCTanaka et al. (2021) focused on web-based comprehension, and ChartQAMasry et al. (2022) specialized in chart-based QA. While these benchmarks made significant contributions, they lacked difficulty grading and cross-domain generalization, due to the limited capabilities of models at the time. With the advent of MLLMs, more complex benchmarks emerged Fu et al. (2024a); Wu et al. (2024); Guan et al. (2024); Li et al. (2023a; 2024); Zhao et al. (2024); Feng et al. (2024), offering a more comprehensive framework for document understanding. OCRBenchV2Fu et al. (2024b) expanded OCR evaluation with various subtasks, while FoxLiu et al. (2024b) improved document parsing through ROI-based methods. OmniDocBenchOuyang et al. (2024) extended evaluation to cross-modal tasks, covering a wider range of parsing challenges. Despite these advancements, these benchmarks still lack explicit difficulty grading, limiting their ability to assess performance across tasks of varying complexity.

**Document Understanding Model**  Optical Character Recognition (OCR) is a core task in computer vision. Existing methods can be broadly categorized into component-based and end-to-end approaches. Component-based methods Wang et al. (2024); Blecher et al. (2023); Du et al. (2021) adopt a modular pipeline that assembles multiple expert-designed components such as layout analysis Zhong et al. (2019), text detection Tian et al. (2016); Liao et al. (2017); Zhou et al. (2017); Liu et al. (2019), region extraction, and content recognition LeCun et al. (1998); Graves et al. (2006); Li et al. (2023b). In contrast, end-to-end OCR models Wei et al. (2024b), especially those driven by MLLMsLiu et al. (2023); Bai et al. (2023); Wei et al. (2023); Ye et al. (2023a); Chen et al. (2024); Liu et al. (2024f;c), aim to unify perception and reasoning within a single architecture. Most MLLM-based OCR systems utilize CLIP Radford et al. (2021) as the vision backbone, while coupling it with a language model to jointly process visual and textual information in a unified framework. Recent works Chen et al. (2024); Liu et al. (2024f); Ye et al. (2023b) adopt sliding-window strategies that partition the image into patches to cope with long and high-resolution inputs like PDFs.

**Continual Learning of LLMs**  Continual learning in large language models (LLMs) faces the core challenge of catastrophic forgetting. Existing approaches mainly use two strategies: data replay Sun et al. (2020); Kanwatchara et al. (2021) and parameter freezing Dou et al. (2024); Razdaibiedina et al. (2023); Liu & Huang (2023). Data replay mitigates forgetting by revisiting prior task samples during new task training. LAMOLSun et al. (2020), for example, generates pseudo-samples to avoid storage-based replay, while other methods like experience replay and interleaved task training integrate old data into the loop. Parameter freezing protects existing knowledge by limiting parameter updates, using techniques such as LoRAHu et al. (2022), adapters Zhang et al. (2022), or prompt tokens Wang et al. (2022). Regularization methods like Elastic Weight Consolidation (EWC) Kirkpatrick et al. (2017) preserve key weights via importance constraints, while newer methods like Task Vector Ilharco et al. (2023) and Gradient Projection Saha et al. (2021) operate at the gradient or representation level to reduce task interference and enhance generalization. In contrast to these approaches, which typically rely on limited pre-training datasets, our method offers a more efficient

and comprehensive alternative. However, directly applying traditional continual learning methods to MLLMs presents significant hurdles: regularization methods like EWC are often computationally prohibitive for billion-parameter models due to the substantial cost of calculating the Fisher Information Matrix, while data replay methods encounter strict storage and privacy limitations. Consequently, Parameter-Efficient Fine-Tuning (PEFT) approaches, particularly LoRA, have become the most widely adopted paradigm for adapting large-scale models under resource constraints.

## 3 BENCHMARK

### 3.1 COMPLEXITY DECOUPLING

As illustrated in Fig. 1, a difficulty-decoupled evaluation protocol named General Document Intelligence Benchmark (GDI-Bench) is introduced to comprehensively assess MLLMs' capabilities in visual information comprehension and reasoning. Distinct from VisualSimpleQAWang et al. (2025)'s structural decomposition of fact-seeking question answering tasks into visual recognition and knowledge dimensions through MLLM model architecture analysis, the proposed framework characterizes task complexity from fundamental difficulty perspectives by decoupling it into visual complexity and reasoning complexity. Notably, knowledge popularity is explicitly subsumed within the reasoning complexity dimension in this formulation.

### 3.1.1 VISUAL COMPLEXITY

The visual complexity dimension is operationalized through a hierarchical categorization of document images into three levels: V0 (plain text), V1 (formal representations), and V2 (explanatory representations). V0 exclusively contains unstructured textual elements such as headings and paragraphs. Multimodal tasks on V0 documents typically achieve satisfactory performance via OCR-LLM pipeline architectures. While V1 and V2 are primarily defined by intrinsic document properties (e.g., structured tables for V1 vs. complex layouts for V2), we utilize model performance for empirical validation. As demonstrated in Fig. 3, a systematic analysis of the OmniDocBench Ouyang et al. (2024) benchmark reveals statistically significant performance gaps. This benchmark covers nine document categories.

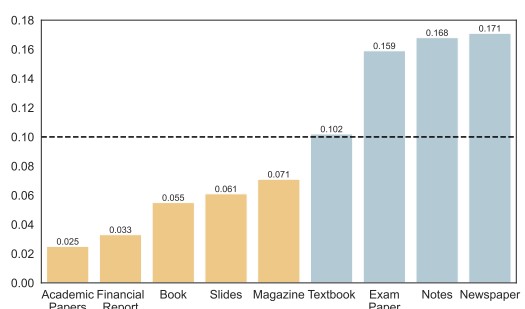

Figure 3: Distribution of visual complexity scores (quantified by the average End-to-End Edit Distance) across nine document categories as reported in the OmniDocBench. The SOTA performance results from the benchmark show a substantial gap between document types, providing empirical justification for our visual complexity taxonomy.

Current pipeline tools and MLLMs show notably worse performance on textbook, exam paper, notes, and newspaper types compared to other forms. These performance drops are directly tied to the inherent characteristics of V2 and show a clear break when the average edit distance reaches 0.10. Based on this empirical evidence, we designate these four challenging categories as V2 and classify the remaining five as V1, ensuring the taxonomy is rooted in document topology rather than solely model errors.

### 3.1.2 REASONING COMPLEXITY

The reasoning complexity characterization is formulated through a behavior-driven taxonomy. Three distinct levels are defined to progressively evaluate document understanding capabilities. It is specifically categorized into R0: Full Page Structured Extract, R1: Information Extract, and R2: Reasoning. Among these, the R0 task in V0 is more similar to OCR, and requires the ability to understand layout information such as tables, formulas, and complex page structures in V1 and V2 type images. For R1 tasks, the model needs not only to recognize visual content but also to understand the task itself in order to accurately extract relevant information. This often involves interpreting lines, tables, labels, and other visual elements present in V1 and V2 images. R2 tasks go a step further by requiring deeper reasoning capabilities, including the comprehension of logical inference and the ability to synthesize information across different modalities or layouts.

## 3.2 ANNOTATION PROCESS

### 3.2.1 DATA SOURCE

For the data collection phase of the GDI-Bench, documents are primarily sourced from OmniDocBench Ouyang et al. (2024), which covers 9 document domains. GDI-Bench is further supplemented with an in-house collection of various document types, including exam papers, reports, newspapers, and so on. This multi-source integration results in a well-rounded and representative dataset that captures the multifaceted nature of real-world document understanding tasks.

### 3.2.2 DATA CONSTRUCTION

As shown in Fig. 4, the data construction process begins with cropping single-layout sub-images from OmniDocBenchOuyang et al. (2024) and inhouse documents to form the V0 raw image set. Based on end-to-end edit distance scores from SOTA models and pipeline tools on OmniDocBench, domains with scores above 0.142 are categorized as V2, indicating high visual complexity. The remaining samples are assigned to V1.

For task construction, R0 tasks are generated using the original annotations or by synthesizing Markdown representations through MinerU Wang et al. (2024). R1 and R2 tasks are created by feeding both Markdown and images into GPT-4o to generate extractive and reasoning QA pairs. Rule-based tasks are designed manually, such as extracting questions from exam papers or retrieving values in colored boxes.

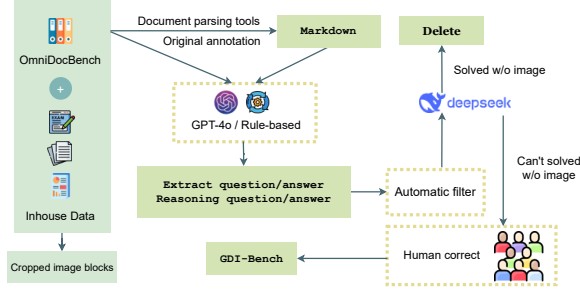

Figure 4: Annotation Process of GDI-Bench.

To ensure data quality, all synthetic cases are first evaluated by models. Cases that can be answered correctly by DeepSeek-R1 Guo et al. (2025) without requiring document input or that are flagged as low quality are filtered out. Finally, a team of PhD-level annotators reviews and verifies the remaining instances to ensure accuracy and correctness. The final GDI-Bench contains a total of 3,660 test cases, distributed across different visual complexity levels (V0, V1, V2) and task types (R0, R1, R2), as summarized in Table 1.

Table 1: Complexity Distribution in the GDI Benchmark.

|       | V0  | V1    | V2    | Total |
|-------|-----|-------|-------|-------|
| R0    | 377 | 521   | 450   | 1,348 |
| R1    | 273 | 498   | 288   | 1,059 |
| R2    | 298 | 641   | 314   | 1,253 |
| Total | 948 | 1,660 | 1,052 | 3,660 |

Table 2: Comparison of GDI-Bench with Other Document Understanding Benchmarks.

| Benchmark        | Scenario | Task | Image | Difficulty Grading |
|------------------|----------|------|-------|--------------------|
| Seed-bench-2-plus | 8        | 1    | 0.6k  | ✗                  |
| MMTab-eval       | 1        | 9    | 23k   | ✗                  |
| MMC              | 1        | 9    | 1.7k  | ✗                  |
| OCRBenchV2       | 31       | 23   | 9.5k  | ✗                  |
| Fox              | 2        | 9    | 0.7k  | ✗                  |
| **GDIBench**     | 9        | 19   | 2.3k  | ✓                  |

## 3.3 EVALUATION METRICS

We adopt different evaluation metrics depending on the task type. For reasoning tasks, most questions are single-choice for consistency. For non-choice tasks, such as document parsing and extraction (mainly R0 and R1), where answers typically appear verbatim in the source document, we use the *Normalized Edit Distance* (NED) Levenshtein et al. (1966), but compute the score as $1 - \text{NED}$ so that higher scores correspond to better performance. Formally, the case score is:

$$
\text{case}_i = \begin{cases} 1 - \text{NED}(\text{prediction}_i, \text{ground truth}_i) & \text{for non-choice tasks} \\ 1 & \text{for choice-based tasks and correct answer} \\ 0 & \text{for choice-based tasks and incorrect answer} \end{cases} \quad (1)
$$

Before computing NED, we apply a text normalization pipeline to minimize the impact of trivial formatting differences. Specifically, we: (i) convert all text to lowercase, (ii) remove common code-block prefixes such as ```json, ```python, ```latex, and ```markdown, (iii) replace whitespace characters (\n, \t), backslashes, and markdown fences (```) with spaces. We also performed controlled robustness checks by introducing small perturbations (e.g., extra spaces, punctuation changes, or case variations) and confirmed that the normalized strings produce stable NED scores.

The overall GDI-Bench Score is the average over all evaluation cases:

$$\text{GDI-Bench Score} = \frac{1}{N} \sum_{i=1}^{N} \text{case}_i \tag{2}$$

where $N$ is the total number of cases. A higher score indicates better overall performance.

### 3.4 COMPARISON WITH EXISTING DOCUMENT INTELLIGENCE BENCHMARKS

As shown in Table 2, most existing document intelligence benchmarks are limited in either scenario coverage, task diversity, or data scale. In contrast, the GDI-Bench offers a balanced combination of 9 diverse document scenarios and 19 representative tasks, based on a curated set of 2.3k images. Notably, it introduces difficulty grading, a feature absent in all other compared benchmarks.

## 4 METHODOLOGY

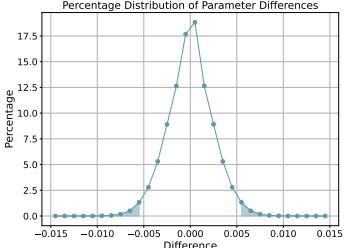

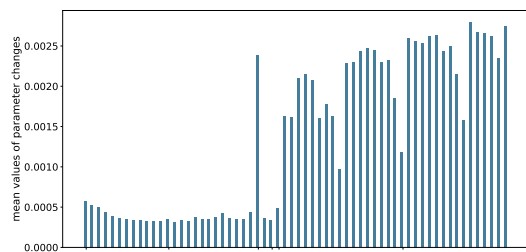

Figure 5: Distribution of parameter differences before and after SFT.

Figure 6: The mean values of parameter changes per layer before and after SFT using 10% of the training set.

We evaluated the InternVL3-8B model on GDI-Bench, conducting a detailed weakness analysis. As shown in Fig. 2, the model performs well on R0 but drops significantly on R1 and R2, highlighting a large performance gap. To validate the ability of GDI-Bench to identify model weaknesses, we aimed to address these gaps in R1 and R2. For this purpose, we constructed a multi-source training dataset that is structurally aligned with the tasks in R1 and R2, but sourced from entirely different data domains, detailed in App. A.2. We performed SFT on the InternVL3-8B model using the constructed training set. However, subsequent evaluations revealed significant catastrophic forgetting, where the model lost important knowledge acquired during pretraining. As shown in Table 5, the Full-Parameter Fine-Tuning Model exhibited degradation in the R0 domain, where it had previously performed well.

To investigate the phenomenon, we comprehensively analyze parameter update dynamics in models subjected to full-parameter supervised fine-tuning. Our experimental results reveal significant sparsity in learned parameter adaptations: over 95% of parameters exhibit minimal changes, while a critical sparse subset (5%) undergoes substantial modification (¿0.005), as depicted in Fig. 5. Inspired by the Lottery Ticket Hypothesis Frankle & Carbin (2018), we integrate this discovery with the hypothesis and formulate our theoretical proposition: For any pre-trained MLLM with parameters $\Theta = \{\theta_i\}_{i=1}^{n}$, there exists a sparse task-salient sub-network $\hat{\Theta} = \{\hat{\theta}_j\}_{j=1}^{m}$ $(m < n)$ and a domain-specific transformation $\psi$, such that $\psi(f_{\hat{\Theta}}(x))$ achieves performance comparable to the full model $f_{\Theta}(x)$ on the target domain $\mathcal{D}$.

$$\forall x \in \mathcal{D}, \ \mathcal{P}(f_{\Theta}(x)) \approx \mathcal{P}(\psi(f_{\hat{\Theta}}(x))) \tag{3}$$

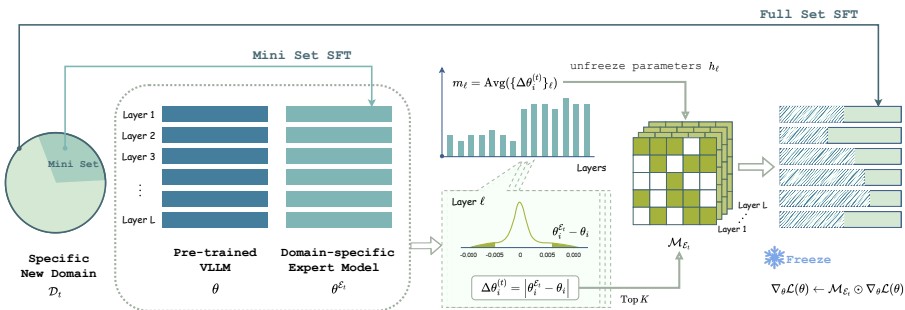

Figure 7: Overview of the Layer-wise Adaptive Freeze-Tuning method.

where $\mathcal{P}(\cdot)$ is the metric for evaluating the performance of the model, and the subnetwork $\hat{\Theta}$ consists of parameters exhibiting high update sensitivity during SFT. Given that only a small subset of parameters exhibit significant task-specific adaptation, we propose performing gradient updates solely on these parameters during SFT, while freezing the majority of remaining parameters to maximally preserve the model's generalization capabilities and foundational knowledge.

We formalize this approach as Layer-wise Adaptive Freeze-Tuning (LW-AFT), which identifies and updates only critical parameters through layer-wise sensitivity analysis. The overview of our method is shown in Fig. 7. For a specific new domain $\mathcal{D}_t$, the expert model (denoted as $\mathcal{E}_t$) is fine-tuned based on the pre-trained parameters $\Theta$ using only a subset $\alpha$ ($\alpha < 1$) of the full dataset. For each domain-specific expert model, the element-wise absolute parameter variation is computed as:

$$\Delta \theta_i^{(t)} = \left| \theta_i^{\mathcal{E}_t} - \theta_i \right|. \tag{4}$$

We conduct a layer-wise analysis of parameter updates $\Delta \theta_i^{(t)}$ before and after SFT. Fig. 6 shows the average magnitude of updates per layer, highlighting different patterns in the vision and language parts. The language layers exhibit a higher degree of parameter modification compared to the visual layers. This observation aligns with the problem localization we performed using GDI-Bench, where the InternVL3-8B model shows a performance degradation in reasoning dimensions. We assume layers with large parameter updates are critical for domain adaptation, as their dynamics closely align with task-specific knowledge transfer. Conversely, layers with minimal updates appear less specialized to specific features, and preserving their stability supports cross-domain generalization.

Therefore, we implement a layer-adaptive freezing parameter allocation strategy. The model is divided into $L$ architectural layers with parameter counts $W = \{w_1, ..., w_L\} \subseteq \mathbb{R}$. Given a global budget of $H$ unfrozen parameters, we employ a proportional allocation mechanism where each layer $\ell$ is allocated $h_\ell$ unfrozen parameters, computed as follows:

$$h_\ell = \frac{m_\ell \cdot w_\ell}{\sum_{i=1}^{L} m_i \cdot w_i} \cdot H \tag{5}$$

where $m_\ell = \mathrm{Avg}(\{\Delta \theta_j^{(t)}\}_\ell)$ is the average absolute change of the $\ell$-th layer.

After determining the number of unfrozen parameters for each layer $\ell$, we first sort the absolute changes $\{\Delta \theta_j^{(t)}\}_\ell$ in descending order, and then select only the top $h_\ell$ parameters for further updates. This selection is performed via the function:

$$\phi_{h_\ell} : \mathbb{R}^{w_\ell} \to \{0,1\}^{w_\ell}, \quad \phi_{h_\ell}\left(\{\Delta \theta_j^{(t)}\}_\ell\right) = \begin{cases} 1, & \text{if } j \in \arg \mathrm{TopK}_k \left(\{\Delta \theta_k^{(t)}\}_\ell, h_\ell\right) \\ 0, & \text{otherwise} \end{cases} \tag{6}$$

The binary masks for the entire network of expert $\mathcal{E}_t$ is $\mathcal{M}_{\mathcal{E}_t} = \left\{ \phi_{h_\ell}\left(\{\Delta \theta_j^{(t)}\}_\ell\right) \right\}_{\ell=1}^{L}$. During the subsequent fine-tuning on a target domain, gradient updates are dynamically masked to exclusively modify parameters corresponding to $\mathcal{M}_{\mathcal{E}_t}$ through:

$$\nabla_\theta \mathcal{L}(\theta) \leftarrow \mathcal{M}_{\mathcal{E}_t} \odot \nabla_\theta \mathcal{L}(\theta), \tag{7}$$

where $\odot$ denotes the Hadamard product. By maintaining the stability of the base network dynamics and avoiding large parameter shifts, LW-AFT effectively mitigates catastrophic forgetting while adapting efficiently to downstream tasks. This approach not only reduces the number of parameters that need to be trained but also preserves global generalization capabilities. With the LW-AFT method, we are able to retain the outstanding performance of the InternVL3-8B model in the R0 domain, while improving its performance in the R1 and R2 domains, as shown in Table 5. This further demonstrates the ability of GDI-Bench to pinpoint model weaknesses.

## 5 EXPERIMENTS

In Section 5.1, we discuss the impact of our proposed Layer-wise Adaptive Freeze-Tuning (LW-AFT) method. In Section 5.2, we perform a comprehensive benchmark evaluation on the GDI-Bench. All experiments are based on the InternVL3-8B model Zhu et al. (2025), with detailed training parameters provided in App. A.1.

### 5.1 LAYER-WISE ADAPTIVE FREEZE-TUNING.

#### 5.1.1 ABLATION STUDY.

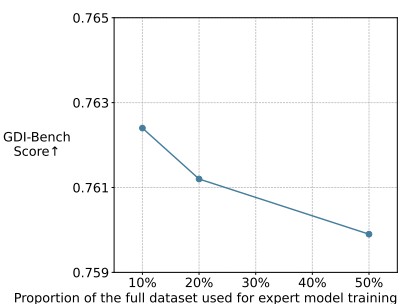 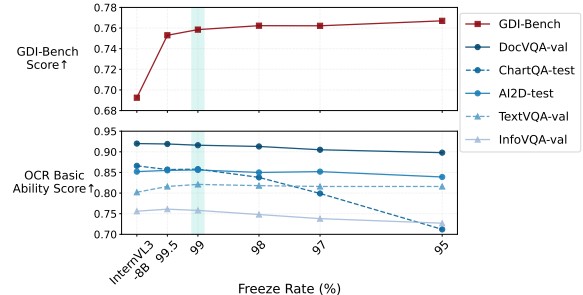

Figure 8: Ablation study on the $\alpha$.     Figure 9: Ablation study on the freeze rate of parameters.

We perform an ablation study on the parameter $\alpha$ in Section 4, which controls the fraction of the dataset used to train domain-specific expert models. In all settings, the LW-AFT method is used with 99% of the model parameters frozen. As shown in Fig. 8, expert models trained with different $\alpha$ values influence the final performance on GDI-Bench. The results indicate that $\alpha = 10\%$ achieves the best performance and data efficiency, and is therefore adopted in our default configuration.

We also study the effect of freezing different proportions of model parameters. Fig. 9 illustrates the performance of the 8B-sized model across GDI-Bench and several other datasets under different freeze ratios. Balancing performance on GDI-Bench and cross-dataset generalization, we find that freezing 99% of parameters is the optimal setting.

#### 5.1.2 EVALUATION OF CATASTROPHIC FORGETTING SEVERITY.

To validate the effectiveness of LW-AFT in mitigating catastrophic forgetting, we evaluate the performance of Full-Parameter Fine-Tuning, LoRA Fine-Tuning, and the LW-AFT method across a range of datasets, including OCR, Chart, Document Understanding, and Multimodal Multilingual Understanding datasets Mathew et al. (2021); Masry et al. (2022); Hiippala et al. (2021); Singh et al. (2019); Liu et al. (2024e); Ouyang et al. (2024); Mathew et al. (2022); Li et al. (2024); Liu et al. (2024d).All fine-tuning experiments were conducted using the dataset detailed in Appendix A.2. Results in Table 3 show that LW-AFT effectively mitigates forgetting after SFT and even surpasses the base model on datasets like TextVQA, InfoVQA, and Seed2plus, confirming the strength of our training strategy.

#### 5.1.3 CROSS-DOMAIN AND CROSS-TASK EVALUATION.

To evaluate LW-AFT's generalization, we design cross-domain and cross-task experiments. For cross-domain evaluation, T1 trains on paragraph beginning extraction in Scientific papers and tests

Table 3: The performance of different training methods on multiple datasets.

| Method | DocVQA↑ | ChartQA↑ | AI2D↑ | | TextVQA↑ | OCRBench↑ | OmniDocBench↓ | InfoVQA↑ | Seed2plus↑ | MMBench↑ | |
|---|---|---|---|---|---|---|---|---|---|---|---|
| | val | test | test | no mask | test | - | - | val | - | en | cn |
| Base | 92.0 | 86.6 | 85.2 | 92.6 | 80.2 | 880 | 0.21 | 75.6 | 69.7 | 85.5 | 85.6 |
| Full-Parameter Fine-Tuning | 39.4 | 33.3 | 43.1 | 49.4 | 22.7 | 228 | 0.42 | 25.9 | 44.8 | 15.6 | 14.6 |
| LoRA Fine-Tuning | 91.3 | **85.8** | 85.4 | 93.1 | 81.6 | 862 | **0.23** | 74.4 | 69.7 | **85.2** | **84.8** |
| LW-AFT (Ours) | **91.6** | 85.8 | **85.6** | **93.2** | **82.1** | **871** | 0.23 | **75.8** | **70.6** | 85.2 | 84.8 |

on the same task in Infograph domain, T2 trains on information extraction and format organization in Exam papers and tests on the same task in Infograph domain, and T3 trains on table-based reasoning QA in Scientific papers and tests on the same task in Financial reports domain. For cross-task evaluation, T4 utilizes the Newspaper domain, where the model is trained on header extraction and format organization, and then tested on fine-grained header field extraction, including publication date, editor, email, and phone number.

Table 4: The performance of different training methods on cross-domain and cross-task scenarios.

| Method | T1 | T2 | T3 | T4 | | | |
|---|---|---|---|---|---|---|---|
| | | | | date | editor | email | phone |
| Base | 0.216 | 0.383 | 0.840 | 0.101 | 0.884 | 0.398 | 0.398 |
| Full-Parameter Fine-Tuning | **0.040** | 0.913 | 0.600 | 0.864 | 0.992 | 0.783 | 0.989 |
| LoRA Fine-Tuning | 0.102 | 0.473 | 0.300 | 0.093 | 0.606 | 0.171 | 0.172 |
| LW-AFT (Ours) | 0.096 | **0.365** | **0.200** | **0.010** | **0.316** | **0.058** | **0.153** |

We compare the performance of Full-Parameter Fine-Tuning, LoRA Fine-Tuning, and the LW-AFT method under both settings, as shown in Table 4, which displays the normalized edit distance for each task. As can be seen, our method demonstrates strong cross-domain and cross-task capabilities, significantly outperforming the LoRA fine-tuning, further demonstrating the effectiveness of the LW-AFT method's generalization ability in the task dimension.

Table 5: The performance of different open-source and closed-source large models and different training methods on GDI-Bench.

| Model | Size | R0V0 | R0V1 | R0V2 | R1V0 | R1V1 | R1V2 | R2V0 | R2V1 | R2V2 | Overall |
|---|---|---|---|---|---|---|---|---|---|---|---|
| Claude-3-5-Sonnet (2024-10-22) | >175B | 0.90 | 0.26 | 0.22 | 0.42 | 0.63 | 0.20 | 0.86 | 0.59 | 0.44 | 0.509 |
| Gemini-2.0-Flash | - | 0.94 | 0.58 | 0.48 | 0.37 | 0.77 | 0.69 | 0.92 | 0.73 | 0.41 | 0.662 |
| GPT-4o (2024-11-20) | ∼200B | 0.96 | 0.60 | 0.43 | 0.39 | 0.71 | 0.57 | 0.81 | 0.51 | 0.35 | 0.593 |
| Grok-2-Vision-on | - | 0.67 | 0.20 | 0.16 | 0.21 | 0.31 | 0.30 | 0.77 | 0.46 | 0.41 | 0.377 |
| o3 (2025-04-16) | - | 0.97 | 0.62 | 0.41 | 0.95 | 0.80 | 0.68 | 0.96 | 0.85 | 0.55 | 0.746 |
| Claude-3-7-Sonnet (2025-02-19) | - | 0.97 | 0.69 | 0.50 | 0.92 | 0.82 | 0.62 | 0.87 | 0.61 | 0.50 | 0.712 |
| Qwen2.5-VL-72B-Instruct | 72B | 0.97 | 0.69 | 0.57 | 0.60 | 0.81 | 0.72 | 0.90 | 0.63 | 0.49 | 0.706 |
| Qwen2.5-VL-7B-Instruct | 7B | 0.92 | 0.39 | 0.31 | 0.81 | 0.58 | 0.32 | 0.82 | 0.56 | 0.43 | 0.562 |
| Qwen3-VL-8B-Instruct | 8B | 0.85 | 0.32 | 0.19 | 0.75 | 0.66 | 0.79 | 0.92 | 0.68 | 0.57 | 0.613 |
| Ovis2-8B | 8B | 0.96 | 0.66 | 0.50 | 0.42 | 0.76 | 0.73 | 0.93 | 0.56 | 0.54 | 0.671 |
| InternVL3-8B | 8B | 0.96 | 0.65 | 0.56 | 0.35 | 0.77 | 0.47 | 0.89 | 0.62 | 0.50 | 0.652 |
| Full-Parameter Fine-Tuning Model | 8B | 0.94 | 0.56 | 0.39 | 0.64 | 0.71 | 0.67 | 0.65 | 0.44 | 0.51 | 0.599 |
| LoRA Fine-Tuning Model | 8B | 0.97 | 0.66 | 0.56 | 0.87 | 0.82 | f0.77 | 0.91 | 0.71 | 0.68 | 0.759 |
| GDI-Model (Ours) | 8B | 0.97 | 0.65 | 0.56 | 0.89 | 0.82 | 0.77 | 0.93 | 0.74 | 0.68 | **0.762** |

## 5.2 BENCHMARK EVALUATION RESULTS.

We employ the LW-AFT method on the complete training dataset, resulting in the development of an 8B-sized GDI-Model. To assess the effectiveness of our GDI-Model on generalized OCR tasks, we compare it against several SOTA MLLMs—namely Qwen2.5VL-7B, Qwen2.5VL-72B Bai et al. (2023), Qwen3VL-8B Yang et al. (2025), Ovis2-8B Brandt et al. (2008), Gemini-2.0-Flash Team et al. (2023), GPT-4o (2024-11-20) Hurst et al. (2024), o3 (2025-04-16) OpenAI (2025), Claude-3-5-Sonnet, Claude-3-7-Sonnet Anthropic (2025), and Grok-2-Vision-on and InternVL3-8B Zhu et al. (2025) on GDI-Bench's multi-level challenge suite (V0-R0 through V2-R2). All evaluations were conducted with the same preprocessing pipeline, as shown in Fig. 2 and Table 5. We also conducted experiments on document-oriented MLLMs, detailed in App.A.7.

A global analysis of Table 5 reveals distinct performance patterns across model scales and task types. Generally, increased visual complexity (from V0 to V2) leads to performance degradation across all evaluated models, highlighting the challenge of interpreting complex layouts and graphical elements. Regarding reasoning complexity, while R2 (Reasoning) scores generally appear higher than R1 (Information Extraction) scores, this is largely due to metric heterogeneity: R2 utilizes Accuracy (with a random baseline of $\sim$0.25), whereas R1 uses 1-NED. Specifically within the R1 domain, we observe a counter-intuitive trend where performance on V1 documents typically exceeds that on V0. This phenomenon is attributable to the sensitivity of the NED metric to output length: V0 tasks often involve extracting short text spans where a single character error results in a high penalty, whereas V1 tasks frequently require extracting tables or lists into long-format JSON outputs, which naturally dilutes the impact of minor errors and leads to higher overall scores. Among the baselines, larger models like Claude-3-7-Sonnet and Qwen2.5-VL-72B demonstrate robust performance, particularly in high-reasoning tasks (R2). However, smaller open-source models (e.g., Qwen2.5-VL-7B, Ovis2-8B) show competitive capabilities in structured extraction (R1V1), suggesting that model size is not the sole determinant of document parsing ability.

Compared to the base InternVL3-8B, our model maintains strong R0-level performance and shows clear gains at R1 and R2, validating the effectiveness of the LW-AFT method. GDI-Model matches Qwen2.5-VL-72B at R0 and outperforms it at higher reasoning levels. At R2, our model rivals Gemini-2.0-Flash under low to medium visual complexity (V0/V1), and even surpasses it under high complexity (V2). These results highlight that, despite a smaller parameter size, our model achieves or exceeds the performance of much larger open- and closed-source counterparts.

## 6 CONCLUSION

We introduce GDI-Bench, a document-domain benchmark with broad coverage and a pioneering difficulty grading system. By decoupling visual and reasoning complexity, it enables systematic evaluation and model optimization guidance. We further introduce LW-AFT (Layer-wise Adaptive Freeze-Tuning) and the corresponding GDI-Model, which reduces catastrophic forgetting during SFT. This method preserves general capabilities while boosting cross-domain and cross-task performance. GDI-Model achieves strong results on GDI-Bench and other benchmarks. We hope that the GDI-Bench can help the advancement of future models and perhaps inspire new theoretical insights.

## 7 ETHICS STATEMENT

GDI-Bench is built from publicly available and curated document sources without personally identifiable or sensitive data. All synthetic cases are automatically filtered and human-verified to ensure quality. The benchmark is intended for academic research; we caution against misuse in high-stakes applications and encourage responsible deployment of document intelligence systems.

## 8 REPRODUCIBILITY STATEMENT

We will release GDI-Bench, including dataset, evaluation scripts, and LW-AFT training code, at `https://huggingface.co/GDIBench`. The training dataset detail is provided in App. A.2. All hyperparameters and training setups are detailed in the App.A.1, and experiments were conducted on InternVL3-8B with 8 A100 GPUs.

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

# A   APPENDIX

## A.1   TRAINING DETAILS AND HYPERPARAMETERS

We use the InternVL3-8B model as the base model and perform supervised fine-tuning on it. In our experiments, both the LW-AFT method and Full-Parameter Fine-Tuning utilize the same set of training hyperparameters, as shown in Table 6. The training process was conducted on 8 A100 GPUs. The specific hyperparameters for LoRA fine-tuning are provided in Table 7.

Table 6: Hyperparameters of the Full-Parameter Fine-Tuning and Layer-wise Adaptive Freeze-Tuning.

| Hyperparameters | Value |
| --- | --- |
| Force image size | 448 |
| Drop path rate | 0.1 |
| Max dynamic patch | 20 |
| Train epochs | 1 |
| Learning rate | 3e-5 |
| Weight decay | 0.05 |
| Warm-up ratio | 0.03 |
| Lr scheduler | cosine |
| Batch size | 12 |
| Max seq length | 8192 |

Table 7: Hyperparameters of the LoRA Fine-Tuning.

| Hyperparameters | Value |
| --- | --- |
| Force image size | 448 |
| Drop path rate | 0.0 |
| Use LLM LoRA | 16 |
| Max dynamic patch | 20 |
| Train epochs | 1 |
| Learning rate | 3e-5 |
| Weight decay | 0.05 |
| Warm-up ratio | 0.03 |
| Lr scheduler | cosine |
| Batch size | 12 |
| Max seq length | 8192 |

## A.2   TRAINING DATA

We construct a training dataset for supervised fine-tuning to enhance the base model InternVL3-8B's capabilities in the R1 and R2 domains. This dataset contains approximately 200,000 QA-form tasks, primarily focused on tasks similar to R1 and R2, but not identical. The data sources are strictly different from those of GDI-Bench. Detailed information on the dataset size and task settings can be found in Table 8.

## A.3   DETAILED DEFINITIONS OF THE 19 SUB-TASKS

To provide a comprehensive understanding of the capability evaluation within GDI-Bench, we explicitly define the 19 sub-tasks employed in our experiments.

- **Exam → JSON**: Parse the hierarchical structure of an exam paper (including questions, options, and sub-questions) and organize the content into a structured JSON format.
- **Color-Box Extraction**: Locate and extract text content specifically contained within colored highlight boxes in the document.

Table 8: Composition of the training set.

| Domain | Task | Training data |
|---|---|---|
| Newspaper | Extract header information of a newspaper page.
Answer reasoning questions for newspaper page (/ multiple choice)...... | 29k |
| Scientific Paper | Extract the author information of the paper.
Answer reasoning questions for a paragraph on a scientific paper page (/ multiple choice)...... | 108k |
| Infograph | Extract the maintitle.
Answer reasoning questions for a infographic page (/ multiple choice)...... | 32k |
| Exam paper | Full text extraction of exampaper pages.
Extract the content of the exam paper based on the type of question and the question number...... | 12k |
| Financial report | Extract the table on a page and convert to LaTeX format.
Answer reasoning questions for a table on a financial report page (/ multiple choice)...... | 3k |
| Handwritten Content | Recognize handwritten content in English and Chinese. | 12k |

- **Table Extraction**: Identify table structures within the document and accurately transcribe them into LaTeX tabular format.

- **Slide QA**: Answer questions requiring understanding of content presented in presentation slides.

- **Exampaper Number Extract**: Retrieve the specific content of a question or section from an exam paper based on a provided question number.

- **Paper Author Extraction**: Extract author names and their corresponding affiliations from academic papers and organize the relationship into a structured JSON object.

- **Single-Table Reasoning**: Perform calculations or logical inferences based on data contained within a single table.

- **Multi-Table Reasoning**: Perform complex reasoning tasks that require synthesizing and comparing information across multiple tables.

- **Paper Table Reasoning**: Perform reasoning tasks based on scientific data tables embedded within academic paper pages.

- **Text Reasoning**: Answer reasoning questions based solely on unstructured plain text content.

- **Doc Reasoning**: Answer reasoning questions that require understanding the global context, complex layout, or cross-modal elements of a general document.

- **Newspaper Date**: Locate and extract the precise publication date string from a newspaper layout.

- **Newspaper Email**: Identify and extract contact email addresses embedded within a newspaper page.

- **Newspaper Tel**: Identify and extract contact telephone numbers embedded within a newspaper page.

- **Chart → Markdown**: Interpret data visualization elements (charts, plots) and convert the underlying data values into a Markdown table representation.

- **Header Extract**: Locate and extract the main headers or titles from the document page.

- **OCR Markdown**: Convert document content, including mathematical formulas, lists, and bold text, into standard Markdown format to preserve logical structure.

- **OCR Text**: Perform basic Optical Character Recognition (OCR) on plain text documents without retaining complex formatting or layout structures.

- **Infograph QA**: Answer questions based on the synthesis of visual and textual information presented in infographics.

## A.4 QUALITATIVE ANALYSIS.

In this section, we provide a qualitative analysis, comparing several cases to demonstrate the performance of the GDI model, as shown in Figures 10-16.

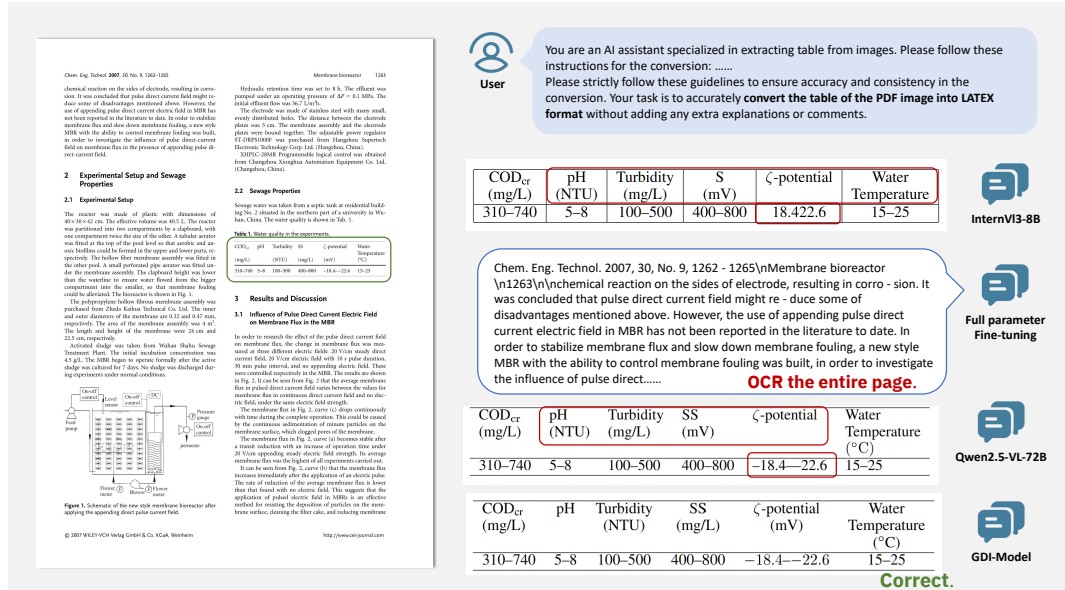

Figure 10: Table extraction task, both the InternVL3-8B model and the Qwen2.5-VL-72B model made errors. The full parameter fine-tuned models failed to handle the table extraction task due to catastrophic forgetting and ended up performing OCR on the entire page's content. In contrast, the GDI-Model successfully extracted the table and output it in LaTeX format.

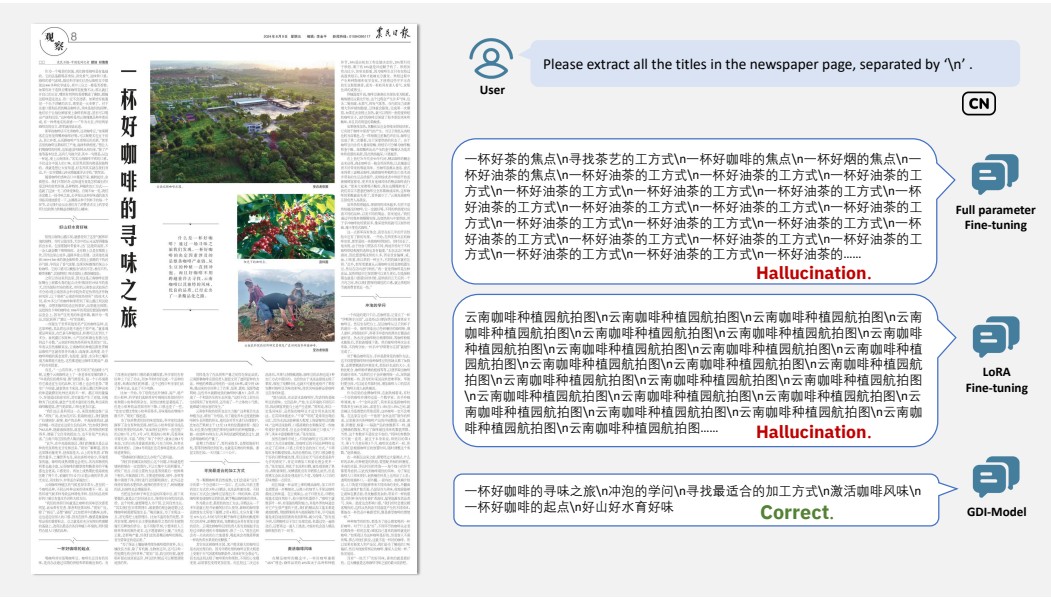

Figure 11: Newspaper titles extraction task, both full parameter fine-tuned models and LoRA fine-tuned models tend to generate model hallucinations, whereas the GDI-Model, obtained using the LW-AFT method, is able to complete the task correctly.

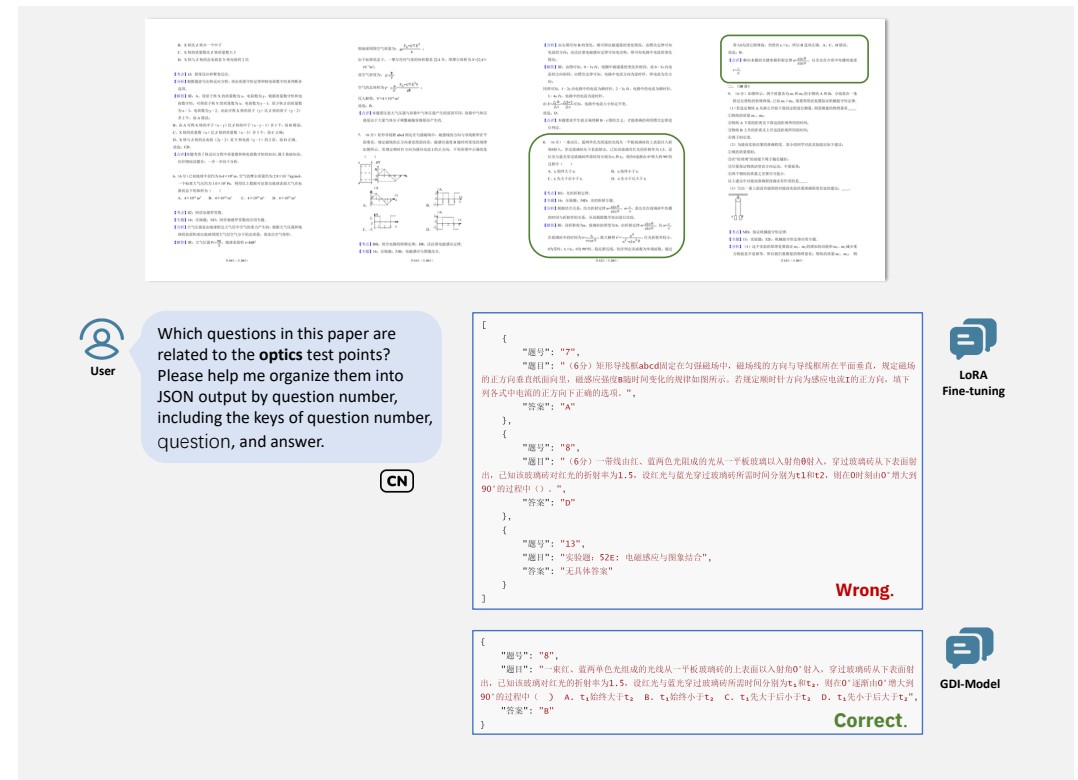

Figure 12: Task of extracting and organizing test points into JSON format, the LoRA model fails to accurately locate the key points or produce correct outputs, whereas the GDI-Model, obtained using the LW-AFT method, is able to complete the task correctly.

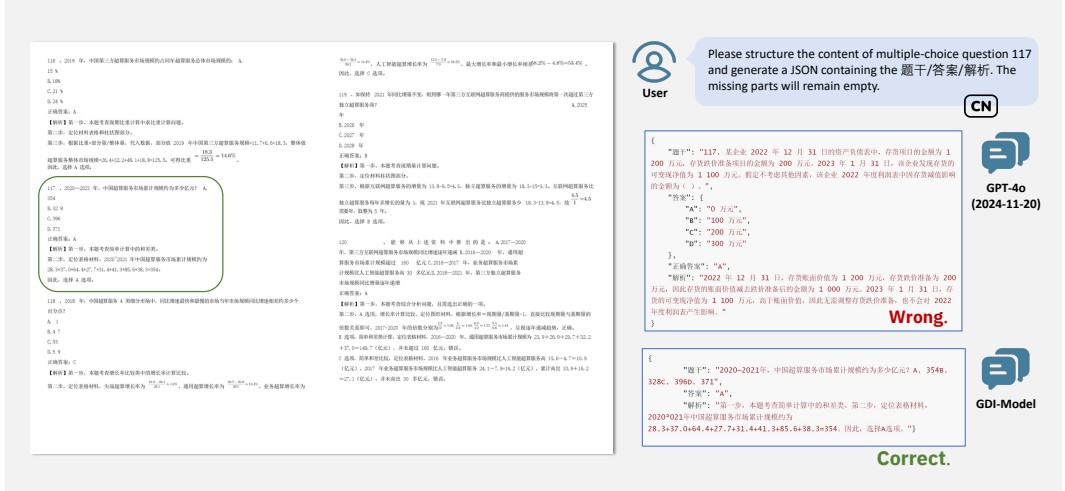

Figure 13: The task of extracting and organizing problems by question number into JSON format, the GPT-4o model fails to correctly locate the questions based on their numbers and ends up fabricating non-existent questions. In contrast, the GDI-Model successfully extracts the questions and outputs them in the specified JSON format.

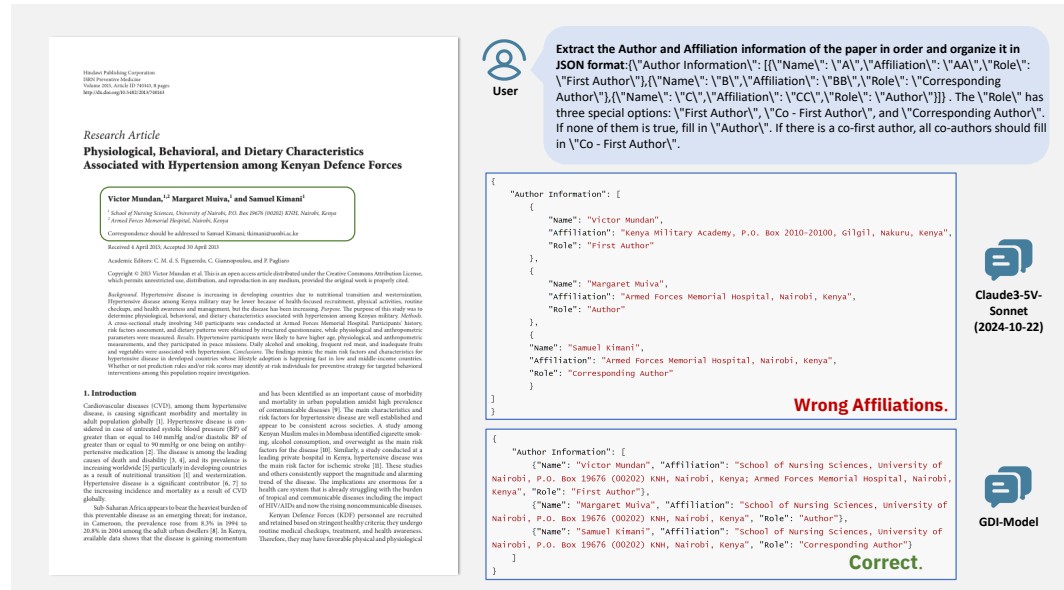

Figure 14: The task of extracting paper author information and organizing it into JSON format, the Claude3-5V-Sonnet model extracted incorrect affiliation information, while the GDI-Model successfully completed the task.

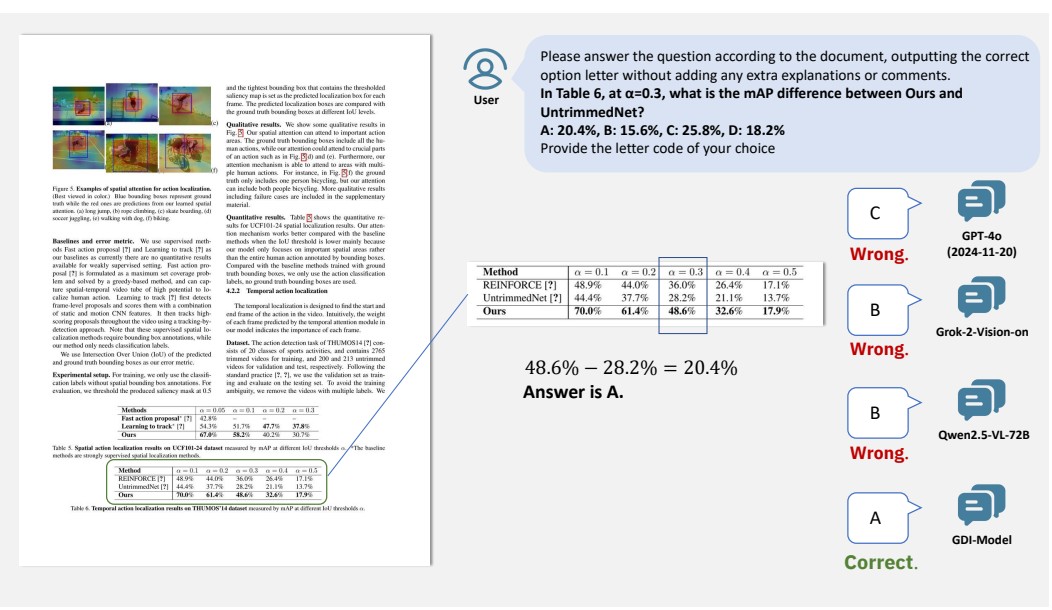

Figure 15: Table reasoning tasks in scientific paper page, the GPT-4o, Grok-2-Vision-on, and Qwen2.5-VL-72B models provided incorrect answers, whereas the GDI-Model answered correctly.

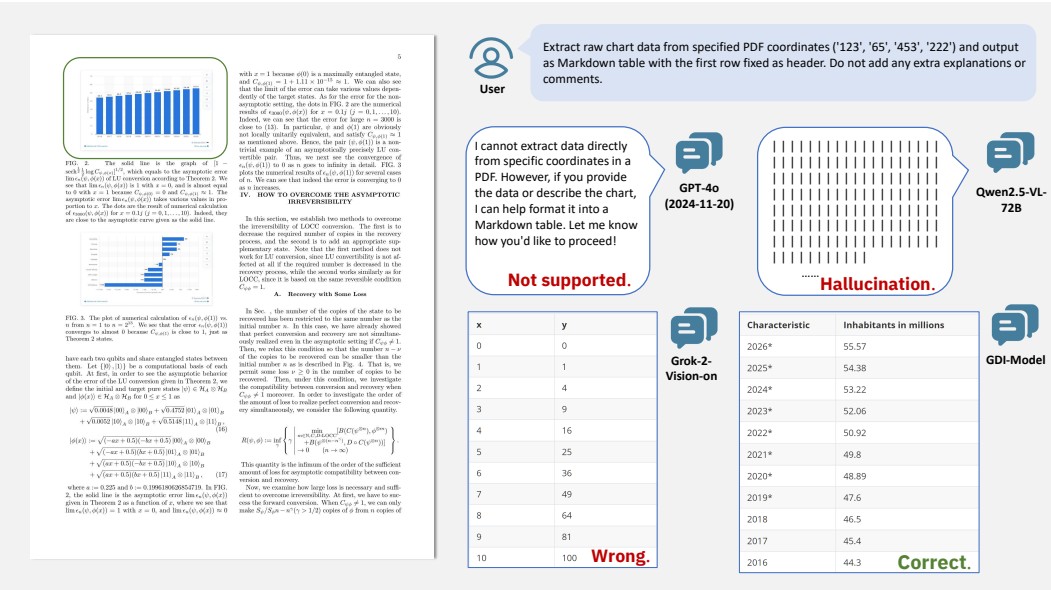

Figure 16: The task of converting chart data within a region into a markdown table, the GPT-4o model does not support the task, Qwen2.5-VL-72B produced a model hallucination, and Grok-2-Vision-on incorrectly converted the data and failed to extract it. In contrast, the GDI-Model completed the task.

## A.5 SAMPLE PROMPTS FOR EVALUATION.

---

**Prompt template for document extraction (v0 tasks)**

Extract all text from the image accurately without interpretation and output only the text without adding any extra explanations or comments.

---

---

**Prompt template for document conversion (v1 & v2 tasks)**

You are an AI assistant specialized in converting PDF images to Markdown format. Please follow these instructions for the conversion:

- **1. Text Processing:**
    - Accurately recognize all text content in the PDF image without guessing or inferring.
    - Convert the recognized text into Markdown format.
    - Maintain the original document structure, including headings, paragraphs, lists, etc.
- **2. Mathematical Formula Processing:**
    - Convert all mathematical formulas to LaTeX format.
    - Enclose inline formulas with \( \). For example: This is an inline formula $E = mc^2$.
    - Enclose block formulas with \[ \]. For example:
$$\frac{-b \pm \sqrt{b^2 - 4ac}}{2a}$$
- **3. Table Processing:**
    - Convert tables to Markdown format.
- **4. Figure Handling:**
    - Ignore figures content in the PDF image. Do not attempt to describe or convert images.
- **5. Output Format:**
    - Ensure the output Markdown document has a clear structure with appropriate line breaks between elements.
    - For complex layouts, try to maintain the original document's structure and format as closely as possible.

Please strictly follow these guidelines to ensure accuracy and consistency in the conversion. Your task is to accurately convert the content of the PDF image into Markdown format without adding any extra explanations or comments.

---

**Prompt template for newspaper date extraction task**

Extract the date information (only including the day, month, and year, if available) of the newspaper, preserving the original punctuation and formatting.

**Prompt template for extracting tables from documents**

You are an AI assistant specialized in extracting table from images. Please follow these instructions for the conversion:

- **1. Text Processing:**
    - Accurately recognize text content in the table without guessing or inferring.
- **2. Mathematical Formula Processing:**
    - Convert all mathematical formulas to LaTeX format.
    - Enclose inline formulas with `\( \)`. For example: This is an inline formula $E = mc^2$.
    - Enclose block formulas with `\[ \]`. For example:
    
    $$\frac{-b \pm \sqrt{b^2 - 4ac}}{2a}$$
    
    .
- **3. Table Processing:**
    - Convert tables to LATEX format.
    - Start with `\begin{tabular}`.
- **4. Output Format:**
    - Ensure the output LATEX has a clear structure with appropriate line breaks between elements.
    - For complex layouts, try to maintain the original document's structure and format as closely as possible.

Please strictly follow these guidelines to ensure accuracy and consistency in the conversion. Your task is to accurately convert the table of the PDF image into LATEX format without adding any extra explanations or comments.

---

**Prompt template for single-choice questions**

Please answer the question according to the document, outputting the correct option letter without adding any extra explanations or comments.

---

**Prompt template for chart extraction from document**

Extract raw chart data from specified PDF coordinates (x1, y1, x2, y2) and output as Markdown table with the first row fixed as header. Do not add any extra explanations or comments.

---

**Prompt template for targeted extraction**

Please extract the text in the yellow color box in the document without adding any extra explanations or comments.

---

## A.6 EVALUATION SETTINGS FOR CLOSED-SOURCE MODELS

This section provides the exact configuration used when evaluating closed-source MLLMs to ensure experimental fairness and reproducibility.

**Prompts.** All prompts used for closed-source models are provided in Appendix A.5. Each API request consists of a system message and the task-specific user prompt. No hidden instructions or additional heuristics were applied.

**Context Length and Max Tokens.** We do not impose external limits on context length. Each request includes the system message, the full text prompt, and the base64-encoded image. The maximum allowable context is determined solely by the model's built-in limits. We leave `max_tokens` at the API default to avoid inconsistencies across different platforms.

**Image Preprocessing.** To ensure input parity, all models receive exactly the same image content. Images are directly encoded as base64 without any resizing, cropping, format normalization, or compression adjustments. No chunking, tiling, or multi-image stitching is applied.

**Evaluation Pipeline.** The evaluation code (included in the submission) invokes each model via a single `chat.completions.create` request containing the full prompt and the base64 image. No retries, resampling, or prompt alterations occur beyond basic exception handling. This ensures strict fairness and consistency across all closed-source baselines.

A.7 EXPERIMENTS ON DOCUMENT-ORIENTED MLLMS

To provide a more comprehensive evaluation, we have expanded the model assessment to include GOT-OCR 2.0 Wei et al. (2024a) and mPLUG-DocOwl2 Hu et al. (2024), offering a broader review of state-of-the-art document-oriented MLLMs. The results are summarized in the table 9.

Table 9: Experiments on document-oriented MLLMs.

| Model | R0V0 | R0V1 | R0V2 | R1V0 | R1V1 | R1V2 | R2V0 | R2V1 | R2V2 | Overall |
|---|---|---|---|---|---|---|---|---|---|---|
| GOT-OCR 2.0 | 0.93 | 0.12 | 0.10 | 0.69 | 0.34 | 0.26 | 0.04 | 0.02 | 0.04 | 0.254 |
| mPLUG-DocOwl2 | 0.31 | 0.03 | 0.03 | 0.42 | 0.16 | 0.31 | 0.53 | 0.53 | 0.25 | 0.224 |
| GDI-Model (Ours) | **0.97** | **0.65** | **0.56** | **0.89** | **0.82** | **0.77** | **0.93** | **0.74** | **0.68** | **0.762** |

Since many document-oriented MLLMs focus primarily on OCR tasks, we selected GOT-OCR as a representative model for testing. GOT-OCR performs well in simpler tasks like R0V0 but struggles to handle more complex extraction and reasoning tasks (R1R2). This highlights their limitation in deeper document understanding, which is crucial for tackling higher-level reasoning. In comparison, the GDI-Model outperforms these models, demonstrating stronger overall performance. This illustrates the value of GDI-Bench's decoupled visual and reasoning grading system.

## A.8 Heatmap Visualization

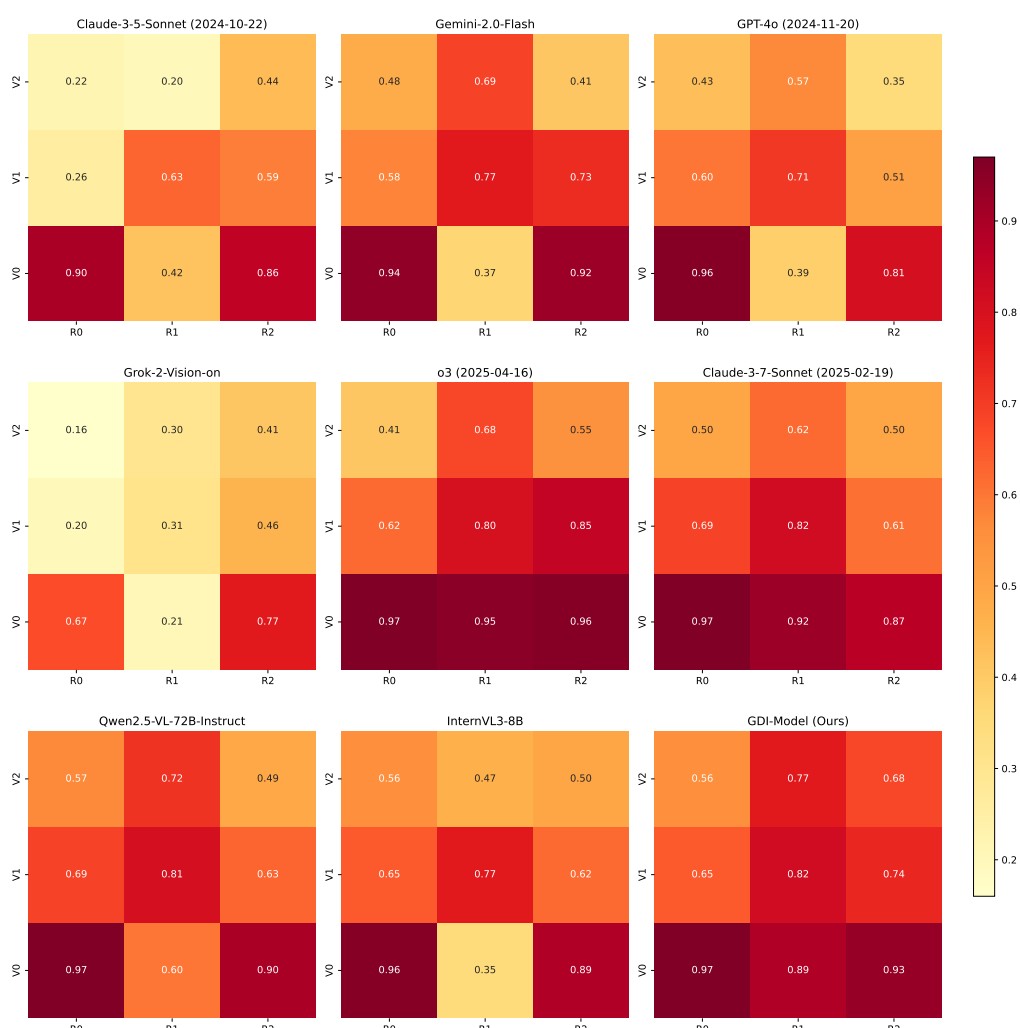

Figure 17: Heatmap visualization of model performance across visual complexity (V0–V2) and reasoning complexity (R0–R2). Each sub-figure corresponds to a specific model, with rows representing visual complexity levels and columns representing reasoning complexity levels. A shared color bar denotes performance values, facilitating cross-model comparison.

To provide a more intuitive illustration of model behavior patterns across complexity levels, we complement the numerical tables in the main paper with heatmap visualizations. As shown in Fig.17, these figures highlight the performance variations along visual complexity (V0–V2) and reasoning complexity (R0–R2) for each evaluated model.

## A.9 Time Consumption for Parameter Selection in LW-AFT

We conducted additional experiments to measure the time consumption required for parameter selection in LW-AFT, specifically for different freezing ratios. The results are shown in Table 10.

As shown in Table 10, the time cost increases as the freeze ratio decreases, which is expected since more parameters are involved in gradient calculation and selection. The overall cost remains acceptable (within 11 minutes even for the 95% freeze ratio), especially considering that this process is performed only once during pre-processing.

Table 10: Time consumption (in seconds) for parameter selection under different freeze ratios in LW-AFT.

| Freeze ratio (%) | 99.5 | 99 | 98 | 97 | 95 |
|---|---|---|---|---|---|
| **Time (s)** | 186.92 | 322.09 | 356.68 | 452.70 | 654.53 |

## A.10 Discussion: Relation to DFT and Catastrophic Forgetting

Recent work such as Dynamic Fine-Tuning (DFT) Wu et al. (2025) revisits the supervised fine-tuning (SFT) objective from a reinforcement learning perspective and proposes a probability-based rescaling of token-level losses to improve the stability and generalization of SFT. We provide a brief discussion here.

First, DFT is primarily designed to address the generalization gap of SFT by rectifying the implicit reward structure induced by cross-entropy training. Its reformulation removes the inverse-probability weighting of expert actions, thereby stabilizing the gradient magnitude and mitigating overfitting to low-probability training tokens. However, DFT is not explicitly motivated by or evaluated on multi-domain or continual-learning scenarios, where catastrophic forgetting is typically observed. Existing evaluations of DFT mainly cover math reasoning, code generation, and multimodal QA, all of which operate within homogeneous task distributions.

Our method mainly focuses on catastrophic forgetting arising in cross-domain, multi-task document understanding. We show that standard SFT significantly degrades previously learned abilities (e.g., R0 tasks) when adapting to new document scenarios. Our proposed Layer-wise Adaptive Freeze-Tuning (LW-AFT) is explicitly designed to preserve domain-general parameters while selectively updating task-sensitive ones, thereby directly addressing forgetting under heterogeneous document distributions.

Given the difference in goals and experimental settings, we regard DFT as conceptually complementary but not directly comparable to LW-AFT for catastrophic forgetting. Although it would be interesting to examine whether DFT's stabilized gradients could partially reduce forgetting, such an investigation is beyond the scope of the current work. We will leave a systematic empirical study of DFT in continual document understanding to future work.

## A.11 Validation of Reasoning Complexity via Text-Only Ablation

To rigorously verify that the R2 reasoning tasks in GDI-Bench necessitate genuine multimodal integration rather than simple textual pattern matching, we conducted a comprehensive "Text-Only" ablation study. We employed a pipeline combining commercial OCR with SOTA LLMs to test the answerability of R2 tasks using exclusively textual input. We evaluated several strong reasoning models, including GPT-4o Hurst et al. (2024), Claude-3.7-Sonnet, and the reasoning-specialized DeepSeek-R1 Guo et al. (2025), comparing their performance against our multimodal GDI-Model.

The quantitative results are presented in Table 11. The analysis reveals a fundamental dependency on visual modality for high-complexity tasks:

- **Semantic Sufficiency in V0:** For plain text documents (V0), the text-only pipelines achieve performance parity with the multimodal GDI-Model (e.g., DeepSeek-R1 reaches 0.93). This confirms that reasoning in V0 scenarios is primarily semantic and can be effectively resolved via advanced text processing, validating the text-centric nature of this complexity level.

- **Visual Necessity in V2:** A drastic performance collapse is observed in tasks involving explanatory representations (V2), such as charts and complex layouts. While the multimodal GDI-Model maintains robust performance (0.68), all text-only baselines degrade significantly. Notably, even the strongest reasoning models like GPT-4o and DeepSeek-R1 drop to scores of 0.18 and 0.35 respectively, often falling near or below the random guessing baseline ($\sim$0.25). This massive performance gap serves as empirical proof that R2 tasks in complex visual scenarios cannot be solved by textual pattern matching alone, strictly requiring the integration of visual layout understanding.

Table 11: Performance comparison of Text-Only (OCR + LLM) pipelines vs. Multimodal GDI-Model on R2 tasks. The significant drop in V2 performance for text-only models highlights the necessity of visual reasoning.

| Model | R2V0 | R2V1 | R2V2 |
|---|---|---|---|
| OCR + GPT-4o | 0.92 | 0.66 | 0.18 |
| OCR + Claude-3-7-Sonnet (2025-02-19) | 0.88 | 0.56 | 0.14 |
| OCR + Qwen2.5-7B-Instruct | 0.79 | 0.49 | 0.27 |
| OCR + Qwen2.5-72B-Instruct | 0.90 | 0.61 | 0.18 |
| OCR + Qwen3-8B | 0.85 | 0.59 | 0.26 |
| OCR + DeepSeek-R1 | 0.93 | 0.70 | 0.35 |
| **GDI-Model (Ours)** | **0.93** | **0.74** | **0.68** |

## A.12 LIMITATIONS

In this section, we outline the limitations of GDI-Bench. Currently, GDI-Bench supports only single-image document understanding tasks, excluding multi-image or multi-document tasks. Future work will expand the benchmark to include these more complex tasks, enhancing its difficulty and enabling wider evaluations.

## A.13 USE OF LLMS

Large Language Models (LLMs) were used to assist with text polishing and improving readability.

