# OpenReview forum: "GDI-Bench: A Benchmark for General Document Intelligence with Vision and Reasoning Decoupling"
_ICLR.cc/2026/Conference — Submitted to ICLR 2026_

### Official Review · Reviewer_84aC · 2025-10-26

**Soundness:** 3
**Presentation:** 3
**Contribution:** 3
**Rating:** 6
**Confidence:** 3

**Summary:**

This paper presents two primary contributions:

GDI-Bench: A new benchmark for document intelligence. Its core innovation is the decoupling of document understanding tasks into two orthogonal difficulty dimensions: visual complexity and reasoning complexity. The benchmark features 2.3k images across 9 key scenarios and 19 document-specific tasks.

GDI-Model: A model developed based on InternVL3-8B , which introduces a training strategy called "Layer-wise Adaptive Freeze-Tuning" (LW-AFT). This strategy aims to address the issue of catastrophic forgetting encountered during SFT on document tasks.

Experimental results show that the 8B-parameter GDI-Model achieves overall performance on GDI-Bench that surpasses larger closed-source models, including GPT-4o, Claude-3-7-Sonnet, and Gemini-2.0-Flash.

**Strengths:**

The paper's core contribution—decoupling visual and reasoning complexity—is well-motivated. The authors correctly point out that an MLLM's failure on a document task could stem from inaccurate visual recognition, limited language organization, or both.

The LW-AFT method is an interesting and novel contribution that addresses a well-known problem in SFT: catastrophic forgetting.

The paper is well-written. Figures like Figure 1 (benchmark framework) and Figure 7 (LW-AFT method) greatly aid in understanding the core ideas.

**Weaknesses:**

The R1 and R2 task QA pairs were generated using GPT-4o. This is a common practice but introduces a potential "LLM-on-LLM" evaluation bias. The benchmark might, to some extent, be measuring the consistency of other models with GPT-4o's output style.

The evaluation of the models is limited. It is necessary to additionally assess Qwen2.5-VL-7B, Qwen3-VL-8B, Ovis2, and others.

I am quite curious about the performance of the recently popular work, DFT, on catastrophic forgetting in the document application domain.

DFT: On the Generalization of SFT: A Reinforcement Learning Perspective with Reward Rectification

**Questions:**

See Weaknesses

---

> ### Author Response · Authors · 2025-11-24
> **Responses to Weaknesses**
>
> **Q1**:
> Thank you for raising this important concern. Although GPT-4o is used to generate initial R1 and R2 QA pairs, we take several steps to minimize any potential “LLM-on-LLM” evaluation bias. Specifically, all GPT-generated QA pairs pass through a multi-stage quality control pipeline. After automated filtering, every remaining instance undergoes human verification by PhD-level annotators with expertise in document intelligence. They independently check each QA pair for factual correctness, clarity, and—crucially—whether the reasoning level matches our task design rather than GPT-4o’s stylistic tendencies. Only QA pairs that meet these criteria are included in the final benchmark. This ensures that the benchmark evaluates models on document-grounded reasoning rather than on matching GPT-4o’s output style. We will clarify this process more explicitly in the revised version.
>
> **Q2**:
>
> Thank you for raising this important concern. In the revised version, we additionally include evaluations of models such as Qwen2.5-VL-7B, Qwen3-VL-8B, and Ovis2. As shown in the table below.
>
> | Model                            | Size       | R0V0     | R0V1     | R0V2     | R1V0     | R1V1     | R1V2     | R2V0     | R2V1     | R2V2     | Overall   |
> | -------------------------------- | ---------- | -------- | -------- | -------- | -------- | -------- | -------- | -------- | -------- | -------- | --------- |
> | Claude-3-5-Sonnet (2024-10-22)   | $>$175B    | 0.90     | 0.26     | 0.22     | 0.42     | 0.63     | 0.20     | 0.86     | 0.59     | 0.44     | 0.509     |
> | Gemini-2.0-Flash     | -          | 0.94     | 0.58     | 0.48     | 0.37     | 0.77     | 0.69     | 0.92     | 0.73     | 0.41     | 0.662     |
> | GPT-4o (2024-11-20)              | $\sim$200B | 0.96     | 0.60     | 0.43     | 0.39     | 0.71     | 0.57     | 0.81     | 0.51     | 0.35     | 0.593     |
> | Grok-2-Vision-on                 | -  | 0.67     | 0.20     | 0.16     | 0.21     | 0.31     | 0.30     | 0.77     | 0.46     | 0.41     | 0.377     |
> | o3 (2025-04-16)                  | -  | 0.97     | 0.62     | 0.41     | 0.95     | 0.80     | 0.68     | 0.96     | 0.85     | 0.55     | 0.746     |
> | Claude-3-7-Sonnet (2025-02-19)   | -  | 0.97     | 0.69     | 0.50     | 0.92     | 0.82     | 0.62     | 0.87     | 0.61     | 0.50     | 0.712     |
> | Qwen2.5-VL-72B-Instruct          | 72B  | 0.97     | 0.69     | 0.57     | 0.60     | 0.81     | 0.72     | 0.90     | 0.63     | 0.49     | 0.706     |
> | **Qwen2.5-VL-7B-Instruct**    | **7B** | **0.92** | **0.39** | **0.31** | **0.81** | **0.58** | **0.32** | **0.82** | **0.56** | **0.43** | **0.562** |
> | **Qwen3-VL-8B-Instruct**   | **8B**     | **0.85** | **0.32** | **0.19** | **0.75** | **0.66** | **0.79** | **0.92** | **0.68** | **0.57** | **0.613** |
> | **Ovis2-8B**    | **8B**| **0.96** | **0.66** | **0.50** | **0.42** | **0.76** | **0.73** | **0.93** | **0.56** | **0.54** | **0.671** |
> | InternVL3-8B | 8B   | 0.96     | 0.65     | 0.56     | 0.35     | 0.77     | 0.47     | 0.89     | 0.62     | 0.50     | 0.652     |
> | Full-Parameter Fine-Tuning Model | 8B         | 0.94     | 0.56     | 0.39     | 0.64     | 0.71     | 0.67     | 0.65     | 0.44     | 0.51     | 0.599     |
> | LoRA Fine-Tuning Model           | 8B         | 0.97     | 0.66     | 0.56     | 0.87     | 0.82     | 0.77     | 0.91     | 0.71     | 0.68     | 0.759     |
> | GDI-Model (Ours)                 | 8B         | 0.97     | 0.65     | 0.56     | 0.89     | 0.82     | 0.77     | 0.93     | 0.74     | 0.68     | 0.762     |
>
> In the revised version, we have added the evaluation results of Qwen2.5-VL-7B, Qwen3-VL-8B, and Ovis2-8b on GDI-Bench to **Table 5**.
>
> **Q3:**
>
> Thank you for the insightful question. We appreciate the reviewer’s interest in the recently proposed Dynamic Fine-Tuning (DFT) method and its potential impact on catastrophic forgetting in the document domain.
>
> Our work primarily focuses on analyzing catastrophic forgetting under standard SFT and proposing LW-AFT as a targeted solution specifically designed for multi-domain document tasks. While DFT introduces a theoretically motivated modification to the SFT objective, its design mainly addresses *generalization* and *stability of token-level updates* rather than multi-domain retention or continual learning. Moreover, DFT has so far been evaluated mostly in math, code, and multimodal reasoning settings, and its behavior under diverse document distributions remains an open question.
>
> Given these differences in problem setting and design goals, we consider an empirical comparison with DFT beyond the scope of our main paper. Nevertheless, we acknowledge the relevance of the reviewer’s point. We will include a short discussion in the appendix  **A.10 DISCUSSION: RELATION TO DFT AND CATASTROPHIC FORGETTING** comparing the conceptual differences between DFT and our LW-AFT, and clarifying why DFT does not directly target catastrophic forgetting in the document domain.

---

### Official Review · Reviewer_paE3 · 2025-10-28

**Soundness:** 3
**Presentation:** 2
**Contribution:** 3
**Rating:** 6
**Confidence:** 3

**Summary:**

This paper presents the General Document Intelligence Benchmark (GDI-Bench), covering 9 key scenarios, 2,300 images, and 19 document-specific tasks. By decoupling visual complexity from reasoning complexity, it creates graded tasks to identify model weaknesses and guide optimization . The study evaluates open-source and closed-source models on GDI-Bench, finding issues like the InternVL3-8B—performing well in the R0 domain but degrading significantly in R1 and R2 . To address these weaknesses and catastrophic forgetting in supervised fine-tuning (SFT), the Layer-wise Adaptive Freeze-Tuning (LW-AFT) method is proposed, leading to GDI-Model. This model achieves state-of-the-art (SOTA) performance on GDI-Bench and other benchmarks.

**Strengths:**

1. GDI-Bench covers 9 key document scenarios, 19 specific tasks, and 2.3k images, ensuring scenario representativeness via multi-source data integration.
2. To address "catastrophic forgetting" in SFT, the study proposes Layer-wise Adaptive Freeze-Tuning (LW-AFT), which improves models’ reasoning performance while preserving basic capabilities.
3. The study pioneers decoupling the complexity of multimodal document understanding into visual complexity (V0-V2) and reasoning complexity (R0-R2), with a difficulty grading mechanism.

**Weaknesses:**

1. GDI-Bench currently only supports single-image document understanding tasks and excludes multi-image or multi-document tasks, limiting its ability to evaluate models in complex real-world scenarios that require cross-image information integration .
2. The proposed Layer-wise Adaptive Freeze-Tuning (LW-AFT) method has mainly been validated on the InternVL3-8B model, with insufficient verification of its generality across models with different architectures (e.g., closed-source models like GPT-4o or other open-source models of varying sizes) .
3. GDI-Bench’s data sources are primarily from OmniDocBench and in-house collections of mainstream domain documents (e.g., scientific papers, newspapers), lacking coverage of tasks in niche domains (e.g., specialized technical manuals, rare language documents), which restricts its adaptability to more diverse practical document processing needs .

**Questions:**

1. How does the Layer-wise Adaptive Freeze-Tuning (LW-AFT) method proposed in the paper alleviate catastrophic forgetting during the supervised fine-tuning (SFT) process?
2.Compared with its base model InternVL3-8B, on which reasoning complexity levels (R0/R1/R2) of GDI-Bench does the GDI-Model show significant performance improvement?

---

> ### Author Response · Authors · 2025-11-24
> **Responses to Weaknesses 1–3 and Questions 1–2**
>
> **W1**:
> Thank you for your comment.  For clarity, the current version of GDI-Bench focuses on single-image document understanding tasks. We appreciate this suggestion, and we plan to extend future versions to include multi-image or multi-document tasks to better reflect real-world document understanding needs.
>
> **W2**:
> We appreciate the your concern regarding the generality of LW-AFT across different model architectures.
>
> First, LW-AFT is a training method that requires access to model parameters and layer-wise representations. Thus, it is inherently infeasible to apply it to fully closed-source models such as GPT-4o, for which neither weights nor training interfaces are available. This limitation applies to all parameter-efficient fine-tuning methods and is not specific to LW-AFT.
>
> To further validate the generality of LW-AFT beyond InternVL3-8B, we additionally conducted experiments on a much smaller model, InternVL2.5-1B, covering diverse expert-domain tasks, including:
>
> - **bbox**: extract content within a given bounding box
> - **color**: extract content within regions matching a given color
> - **start**: extract a full text segment beginning with a given prefix
> - **title**: extract all titles from document pages
> - **handwrite cn/en**: recognize handwritten Chinese and English
>
> The results are summarized below:
>
> | Model (NED↓)             | bbox  | color | start | title | Hand cn | Hand en |
> | ------------------------ | ----- | ----- | ----- | ----- | ------- | ------- |
> | InternVL-2.5-1B          | 0.869 | 0.786 | 0.750 | 0.582 | 0.241   | 0.344   |
> | InternVL-2.5-1B (SFT)    | 0.087 | 0.068 | 0.046 | 0.076 | 0.0157  | 0.0499  |
> | InternVL-2.5-1B (LW-AFT) | 0.181 | 0.097 | 0.066 | 0.075 | 0.0344  | 0.0303  |
>
> We also evaluated catastrophic forgetting on a diverse set of benchmarks:
>
> | Model                    | docvqa-val | chartvqa-human | chartvqa-augmented | mmvet | mmbench-en-dev | mmbench-cn-dev |
> | ------------------------ | ---------- | -------------- | ------------------ | ----- | -------------- | -------------- |
> | InternVL-2.5-1B          | 0.832      | 0.5888         | 0.9328             | 52.5  | 0.717          | 0.646          |
> | InternVL-2.5-1B (SFT)    | 0.0639     | 0.0144         | 0.0208             | 8.2   | 0.013          | 0.013          |
> | InternVL-2.5-1B (LW-AFT) | 0.502      | 0.244          | 0.3968             | 21.7  | 0.570          | 0.488          |
>
>  Across both InternVL3-8B and InternVL2.5-1B, LW-AFT consistently boosts expert-domain performance while retaining a substantial portion of the base model’s general capabilities. These results support that LW-AFT is model-agnostic and scales to architectures and model sizes beyond those reported in the main paper.
>
> **W3**
>
> Thank you for the thoughtful comment. The current GDI-Bench includes 9 scenarios, 1,347 Chinese QA pairs, and 2,313 English QA pairs, all rigorously **human-validated** for accuracy and consistency. Incorporating specialized technical manuals or rare-language documents indeed requires domain expertise and careful calibration, which we currently lack the resources to support. We appreciate this suggestion and agree that broader domain coverage would further strengthen the benchmark. In future versions, we plan to explore adding expert-curated technical manuals and rare-language documents to continue enriching and expanding GDI-Bench’s evaluation scope.
>
> **Q1**:
>
> Thank you for the comment. Our Layer-wise Adaptive Freeze-Tuning (LW-AFT) method is designed to reduce catastrophic forgetting by freezing most model parameters and updating only a small set of important ones found through layer-wise sensitivity analysis. In practice, we see that less than 5% of parameters change during SFT. LW-AFT uses this fact by (i) training a small expert model to estimate which parameters are most sensitive, (ii) allocating a proportion of parameters for each layer based on its sensitivity and selecting the most responsive parameters within that layer, and (iii) updating only this small group while keeping the rest fixed. This helps the model keep its original abilities—especially on R0 tasks—while still adapting well to harder R1/R2 tasks, which greatly reduces catastrophic forgetting.
>
> **Q2**:
>
> Thank you for the comment. As shown in Table 5, the GDI-Model achieves clear and consistent improvements on both R1 and R2 reasoning levels across all visual complexity settings. The detailed numbers are as follows:
>
> - R1:
>   - R1-V0: **0.35 → 0.89**
>   - R1-V1: **0.77 → 0.82**
>   - R1-V2: **0.47 → 0.77**
> - R2:
>   - R2-V0: **0.89 → 0.93**
>   - R2-V1: **0.62 → 0.74**
>   - R2-V2: **0.50 → 0.68**
>
> These gains confirm that LW-AFT successfully boosts the model’s weakest reasoning dimensions (R1/R2) while preserving its strong R0 performance, demonstrating both targeted improvement and effective knowledge retention.

---

### Official Review · Reviewer_HDHb · 2025-10-31

**Soundness:** 1
**Presentation:** 2
**Contribution:** 2
**Rating:** 2
**Confidence:** 4

**Summary:**

The paper describes two different and independent contributions. On one hand, the creation of a new benchmark for document understanding, where the main novelty is the categorization of documents and tasks across two different axes: visual complexity of document and reasoning difficulty of the tasks. This can allow to analyze the performance of the evaluated models according to this two variables. On the other hand, the proposal of a continual learning strategy to mitigate catastrophic forgetting when training the model for different tasks and domains of documents. Both contributions are partially related through the application of the continual learning approach to the training of different tasks in the proposed GDI_Bench.

**Strengths:**

- The paper introduces a new benchmark for document understanding that includes diverse types of documents and reasoning tasks, that can be useful to advance in the research of generic document understanding models, able to perform well across different types of documents and tasks.
- The paper analyzes the problem of catastrophic forgetting in the context of learning differnet tasks for document understandin and proposes a training procedure to overcome this problem.

**Weaknesses:**

- The categorization of documents using visual complexity and reasoning difficulty seems to abstract. It is difficult to know exactly what documents and tasks are behind categories V0,v1 and V2, and R0, R1, R2. I think it is better a categorization based on more focused characteristics of documents and tasks, as it is already done in other existing benchmarks. For instance, categorizing by the type of visual evidences contained in a document or required to answer the question, or by the type of reasoning required (single-page or cross-page, multi-hop reasoning, arithmetic operation, ... )
- Visual complexity is defined according to the performance of current models in OCR conversion, not to the intrinsic characteristics of the documents. OCR performance depends on current benchmarks and models, and thus, it might not reflect actual complexity of documents, but complexity of the documents in that specific benchamark and the ability of models to deal with those types of documents. For instance, in principle, it would seem that slides or magazines can contain more visual complexity than newspapers, but slides have much better score than newspapers.
- In general, the definition of the tasks for each level of difficulty is not clear, especially for R1 and R2. According to table 2, there are 19 different tasks. Which are these 19 different tasks? How are they distributed accross the different levels of difficulty. In addition, R2 tasks are defined as multi-choice QAs. Maybe it is not the best way to evaluate complex reasoning abilities of the models, since a multi-choice question is giving some hints to answer the model. It seems more appropiate to have open QAs
- Tasks in different levels of difficulty have different evaluation metrics and thus, results are not comparable across levels of difficulty. For instance, one would expect that results for R2 tasks are worse than for R1 and R0 tasks, but according to figure 2, they are better. Also results for V1 documents in R1 tasks are better than results for V0 documents which, in principle are less complex. Some discussion on this effect would be necessary.
- The results shown in figure 2 are not discussed in the text of the paper. Only results for InternVL are discussed in section 4. A global joint discussion of these results for all models across different types of documents and tasks would be necessary.
- The initial hypothesis that motivates the proposal of the fine-tuning method in secion 4 is not fully justified. According to table 5, fine-tuning with the full model not only degrades performance for R0 tasks, but also for R2 tasks, that are supposed to be included in the training set. This would suggest that the only explanation for the degradation of performance would not be catastrophic forgetting.
- The proposed fine-tuning approach is only compared with LoRA (with very similar results). A better comparison with other SoA for continual learning would be necessary.
- Section on related work focuses the analysis of document understanding on OCR tasks. Document understanding implies much more than simple OCR.

**Questions:**

- In terms of paper format, figures and tables should be located in the document closer to the section of the text where they are discussed. For instance, figure 2 and table 5 are far away from the text where they are cited.
- How the combination of the results across different models in OmniDocBench is done to get the score that is used to decide the category for visual complexity? How thresholds for V1 and V2 are determined?
- It is not clear what figure 3 exactly shows. From the context I guess that it shows average edit distance score for OmniDocBenc for each type of documents, but more details are needed in the text in order to better explain it.
- What is the training set used to obtain the results in table 3? It is the combination of all training sets for all datasets?

---

> ### Author Response · Authors · 2025-11-24
> **Responses to Weaknesses 1–2**
>
> **W1**:
>
> We appreciate the reviewer’s feedback on our categorization taxonomy. We clarify that our definitions of Visual Complexity (V0-V2) and Reasoning Complexity (R0–R2) are not abstract concepts, but are explicitly grounded in the concrete “visual evidences’’ and “reasoning types’’ mentioned in the comment. As detailed in **3.1 COMPLEXITY DECOUPLING**, visual complexity reflects the graphical structures required for answering: V0  includes plain text, V1 covers tables/lists/equations, and V2 captures charts, figures, and complex layouts. Likewise, the reasoning dimension spans structural perception (R0), direct information extraction (R1), and deeper logical or multi-hop inference (R2). Thus, the taxonomy covers the behaviors the reviewer mentioned, but arranges them in a hierarchical way to reflect increasing levels of difficulty.
>
> We invite the reviewer to visit our open-sourced benchmark to see the full set of tasks and document types covered in each category. **Unlike existing benchmarks that categorize samples purely by document or task features, our taxonomy is designed around the core principle of decoupling visual complexity from reasoning complexity.** A task-based taxonomy would obscure this separation and weaken the benchmark’s ability to diagnose model bottlenecks. In contrast, by explicitly distinguishing visual complexity from reasoning complexity, GDI-Bench enables precise attribution of model errors—revealing whether failures arise from perception or from reasoning.
>
> **W2**:
>
> We thank the reviewer for the comment. We respectfully clarify that our definition of Visual Complexity (V0, V1, V2) is fundamentally established based on the intrinsic characteristics of document elements, rather than solely on model performance scores. As defined in 3.1.1 VISUAL COMPLEXITY, the categories are structured by content type: V0 consists of unstructured plain text; V1 comprises formal representations such as tables, lists, and equations; and V2 involves explanatory representations such as charts, figures, and complex layouts. The performance data referenced in Figure 3 was utilized as a data-driven validation to confirm that these intrinsic groupings indeed correspond to distinct levels of difficulty for current MLLMs, justifying our taxonomy. We have further emphasized this explanation in **3.1.1 VISUAL COMPLEXITY** of the revised version.
>
> P.S.
>
> Regarding the specific comparison between **slides** and **newspapers**, we argue that "Visual Complexity" in the context of document intelligence specifically refers to the complexity of layout analysis and information topology, rather than merely visual richness or color. While slides may appear visually rich, they often feature spatially distinct, sparse blocks of content that are topologically simpler to parse. In contrast, newspapers are intrinsically characterized by dense, multi-column layouts, articles that wrap around images, and varied reading orders. These are objective, intrinsic structural features that create high visual complexity for layout parsing. This structural density explains why newspapers are categorized as challenging and why models objectively struggle with them compared to the often cleaner structure of slides.

---

> ### Author Response · Authors · 2025-11-24
> **Responses to Weaknesses 3–4**
>
> **W3**:
>
> We appreciate the reviewer’s request for greater granularity regarding the task definitions. To clarify, the **19 specific tasks** mentioned in Table 2 are defined as follows:
>
> * **OCR Text**: Perform basic OCR on plain text documents without retaining complex formatting.
> * **OCR Markdown**: Perform OCR to convert document content, including formulas and lists, into Markdown format.
> * **Table Extraction**: Extract tables from the document and output them in LaTeX format.
> * **Header Extract**: Extract headers or titles from the document.
> * **Chart→ Markdown**: Convert chart data or visual representations into a Markdown format.
> * **Exam→ JSON**: Organize the exam paper content into a structured JSON format.
> * **Color-Box Extraction**: Extract content specifically from colored boxes within the document.
> * **Exampaper Number Extract**: Extract specific content from an exam paper based on question numbers.
> * **Paper Author Extraction**: Extract the author and affiliation information from research papers and organize it into JSON format.
> * **Newspaper Date**: Extract the specific date information from a newspaper page.
> * **Newspaper Email**: Extract contact email addresses from a newspaper page.
> * **Newspaper Tel**: Extract contact telephone numbers from a newspaper page.
> * **Text Reasoning**: Answer reasoning questions based on pure text content.
> * **Slide QA**: Answer questions based on the content of presentation slides.
> * **Infograph QA**: Answer questions based on the content of infographics.
> * **Single-Table Reasoning**: Perform reasoning tasks based on a single table.
> * **Multi-Table Reasoning**: Perform reasoning tasks that require synthesizing information across multiple tables.
> * **Paper Table Reasoning**: Perform reasoning tasks based on tables found within academic paper pages.
> * **Doc Reasoning**: Answer reasoning questions based on general document content involving complex layouts.
>
> In the revised version, we have added a listing of these sub-tasks in Appendix **A.3 DETAILED DEFINITIONS OF THE 19 SUB-TASKS**.
>
> For R2 reasoning tasks, we use multiple-choice questions because they provide clear and reliable evaluation. For perception and extraction tasks (such as *OCR Markdown*, *Newspaper Email*, *Table Extraction*), we use open-ended outputs with Normalized Edit Distance, since the model must reproduce exact text without any hints. However, for complex reasoning tasks (like *Multi-Table Reasoning* or *Document Reasoning*), open-ended answers are hard to grade automatically and often cause ambiguity. MCQs avoid this issue by giving a deterministic accuracy metric. To reduce the “hint” problem, we design plausible distractors, such as close numerical values, so the model must actually perform the correct reasoning rather than guess. Overall, this mixed design—open extraction for perception and MCQs for high-level reasoning—keeps the benchmark both realistic and easy to evaluate fairly.
>
> **W4**:
>
> We sincerely thank the reviewer for pointing out the subtle points involved in understanding the performance trends across different difficulty levels. We fully agree that the results across the reasoning levels (R0–R2) cannot be directly compared because the evaluation metrics used for them are different. Specifically, R0 and R1 tasks are evaluated using strictly penalized Normalized Edit Distance (1-NED) for generative extraction, whereas R2 tasks utilize Accuracy for multiple-choice questions. To address this difference and provide a proper baseline for comparison, we included a gray line in the R2 subfig of Figure 2 to denote the expected performance of random guessing (approximately 0.25). Consequently, while the absolute scores for R2 appear higher than those for R1, they must be interpreted relative to this chance-level baseline, in contrast to the extraction tasks, where the effective baseline is near zero.
>
> Regarding the observation that V1 (Formal Representations) results are significantly better than V0 (Plain Text) in R1 extraction tasks,  our analysis shows that this difference mainly comes from the NED metric being very sensitive to the length of the output. In our GDI-Bench, R1 tasks on V0 documents often involve extracting short, specific entities, where a single character error results in a high edit distance ratio, leading to a substantial drop in the 1-NED score. Conversely, R1 tasks on V1 documents frequently involve extracting full tables or lists into structured formats like JSON. These outputs are typically much longer; therefore, minor character-level errors are diluted over the entire sequence, resulting in a comparatively lower NED and a higher overall score. We invite the reviewer to examine the specific cases in our open-sourced benchmark, which illustrate how these varying output lengths inherently influence the evaluation scores despite the visual simplicity of V0 documents. We have added this explanation to **5.2 BENCHMARK EVALUATION RESULTS** in the revised version.

---

> ### Author Response · Authors · 2025-11-24
> **Responses to Weaknesses 5–7**
>
> **W5**:
>
> We thank the reviewer for this suggestion. Regarding Figure 2, it is a visualization of the data presented in Table 5; therefore, we discuss it together with Table 5 in  Section 5. Besides, we note that Section 4 Methodology is dedicated exclusively to the theoretical formulation of our Layer-wise Adaptive Freeze-Tuning (LW-AFT) strategy and does not contain experimental results. We believe the reviewer is referring to the experimental analysis presented in Section 5. We fully agree with the reviewer’s suggestion to provide a more global, joint analysis of all models’ results across different document types and tasks.
> In response to this recommendation, we have revised **Section 5.2 BENCHMARK EVALUATION RESULTS** to include a global joint discussion of the results shown in Figure 2. This expanded analysis moves beyond the specific focus on the InternVL base model to provide a comparative evaluation of all open-source and closed-source models—such as GPT-4o, Claude-3-5-Sonnet, and Gemini-2.0-Flash—across the various document types (V0–V2) and task complexities (R0–R2).
>
> **W6**:
>
> We appreciate the reviewer’s critical assessment regarding the motivation of our method. We respectfully propose that the performance degradation observed in R2 tasks under Full-Parameter Fine-Tuning is indeed a manifestation of **catastrophic forgetting**, specifically the loss of **generalization capabilities** acquired during pre-training. In the context of Multimodal Large Language Models, forgetting is not limited to erasing memory of specific old tasks (like R0 extraction); it also entails the erosion of the robust, general-purpose visual-linguistic alignment and reasoning logic established during the massive pre-training phase.
>
> Although our training data includes a small portion of light reasoning tasks, their composition is fundamentally different in origin from the R2 tasks in the benchmark, and their quantity is limited. As a result, full-parameter fine-tuning still leads the model to **overfit** the specific distribution, templates, and domains of the SFT data, without genuinely improving its capability on R2 tasks. This "over-optimization" for the limited training set destroys the delicate feature manifold required for transferring reasoning skills to the diverse and unseen scenarios in GDI-Bench, leading to the observed degradation in R2 scores (e.g., R2V0 dropping from 0.89 to 0.65). This hypothesis is empirically validated by the success of our proposed LW-AFT method. By freezing the majority of the pre-trained parameters, LW-AFT forces the model to retain its foundational general capabilities while adapting only a sparse set of task-salient parameters. The fact that LW-AFT significantly recovers R2 performance (achieving 0.762 overall compared to FPFT's 0.599) serves as strong evidence that preserving pre-trained representations—i.e., mitigating catastrophic forgetting—is indeed the key to preventing performance degradation in high-level reasoning tasks.
>
> **W7**:
>
> We thank the reviewer for this comment. While we acknowledge that the aggregated performance on general benchmarks appears comparable between LoRA and our method, we respectfully direct the reviewer’s attention to **Table 4**, which reveals a substantial performance divergence in more challenging cross-domain and cross-task scenarios. Specifically, in the Newspaper header extraction task (T4), LoRA exhibits severe catastrophic forgetting with high NED (e.g., 0.606 for editor extraction), indicating a failure to generalize. In contrast, our LW-AFT method maintains robust performance with a significantly lower error rate of 0.316 on the same task. This shows that LW-AFT has a clear advantage in avoiding task interference and keeping strong generalization in complex transfer settings—an important benefit that the averaged metrics in Table 5 do not fully reflect.
>
> Regarding the choice of baselines, we selected LoRA as the main comparator because it is the most widely used and practical method for adapting large MLLMs (8B+ parameters) when resources are limited. Traditional Continual Learning methods like EWC are usually too expensive to run at this scale because computing the Fisher Information Matrix is very costly. Data replay methods also struggle due to large storage demands and potential privacy issues. Consequently, benchmarking against LoRA and Full-Parameter Fine-Tuning provides the most rigorous and practically relevant evaluation for large-scale multimodal adaptation. We have updated the **2 RELATED WORKS** section in the revised version to clearly explain the rationale behind this choice.

---

> ### Author Response · Authors · 2025-11-24
> **Responses to Weaknesses 8 and Questions 1–2**
>
> **W8**:
>
> We thank the reviewer for this comment. However, we would like to clarify that our discussion in **Section 2 RELATED WORKS** is designed to cover the full scope of document intelligence, not just basic text recognition.
>
> First, in the **"Document Benchmark"** subsection, we explicitly discuss a wide range of non-OCR tasks. As detailed in the text, we cover Industrial Document QA (DocVQA), Chart-based QA (ChartQA), Web Page Comprehension (VisualMRC), and ROI-based Document Parsing (Fox, OmniDocBench). These benchmarks focus on complex reasoning, layout analysis, and information extraction, demonstrating that our scope is by no means limited to simple OCR.
>
> Second, regarding the **"Document Understanding Model"** subsection, we extensively discussed OCR-related architectures because many current state-of-the-art document-oriented MLLMs (e.g. GOT-OCR 2.0) are explicitly designed under the paradigm of "General OCR". These models serve as the foundational backbones for the broader document understanding capabilities (R1/R2) evaluated in our paper. Therefore, referencing these models is necessary to contextualize the current technological landscape.
>
> We suspect there might be a misunderstanding where the specific focus of the **"Document Understanding Model"** subsection was perceived as representing the entire Related Work section. We respectfully invite the reviewer to revisit the **"Document Benchmark"** subsection, which substantiates our commitment to a comprehensive analysis of document understanding beyond OCR. We hope this clarification resolves the concern.
>
> **Q1**:
>
> We appreciate the reviewer’s attention to detail regarding the manuscript's presentation. We generally agree with the principle that figures and tables should be located as close as possible to their discussion in the text to ensure a smooth reading experience. However, we would like to offer a clarification regarding the specific placement of the mentioned items:
>
> - **Figure 2:** This figure is first cited in Section **1 INTRODUCTION**, at the beginning of the third paragraph on Page 1. To provide readers with an early visual overview of the model performances as they are introduced, we positioned Figure 2 on Page 3.
> - **Table 5:** This table is currently located on Page 9, which is intentionally placed alongside **5.2 BENCHMARK EVALUATION RESULTS**. This ensures that the detailed numerical results are presented exactly where the comprehensive discussion and comparative analysis of all models take place.
>
> In the final version, we will carefully review the layout to further optimize the proximity between all floats and their primary textual references.
>
> **Q2**:
>
> We thank the reviewer for asking for this clarification regarding our complexity definition methodology.
>
> For  the calculation of Visual Complexity scores, the "score" used to analyze visual complexity is derived from the **average End-to-End Edit Distance** achieved by representative SOTA models and pipeline tools on the OmniDocBench dataset. We aggregated the performance of these models across the 9 distinct document domains provided in OmniDocBench. Since Edit Distance measures the divergence between the prediction and the ground truth, a higher score indicates a higher error rate.
>
> For the determination of thresholds for V1 and V2, we wish to clarify that our visual complexity taxonomy is primarily defined by **intrinsic document properties** rather than solely by model behavior, although model performance was used to empirically validate and operationalize the difficulty grading. The categories are defined as follows:
>
> * **V1 (Formal Representations):** Defined by the presence of structured elements like tables, lists, and equations.
> * **V2 (Explanatory Representations):** Characterized by complex visual features, including multi-column layouts, charts, figures, and flows.
>
> The performance gaps observed in OmniDocBench (illustrated in **Figure 3**) were utilized as evidence to show that these structural differences match real difficulty levels for current models. Specifically, we observed a distinct performance gap where domains possessing V2 characteristics consistently resulted in higher average edit distances compared to V1 domains. Based on this clear empirical break, we set **0.10** as the cutoff threshold to define this distinction. Finally, to avoid overfitting and ensure the difficulty grading aligns with human judgment, a team of PhD-level annotators manually checked and confirmed all instances and their assigned categories. We have further emphasized this explanation in **3.1.1 VISUAL COMPLEXITY** of the revised version.

---

> ### Author Response · Authors · 2025-11-24
> **Responses to Questions 3–4.**
>
> **Q3**:
>
> We apologize for the unclear description in the figure. Figure 3 illustrates the average End-to-End Edit Distance scores derived from an aggregation of SOTA models and pipeline tools evaluated on the OmniDocBench dataset. In the revised version, we have updated the caption of **Figure 3** to clarify this point.
>
> **Q4**:
>
> We appreciate the reviewer for seeking clarification on the experimental setup. We wish to clarify that all comparative methods in Table 3 (Full-Parameter Fine-Tuning, LoRA, and our LW-AFT) were fine-tuned using the specific multi-source training dataset described in Appendix **A.2 TRAINING DATA (Table 8)**. This constructed dataset consists of approximately 200,000 QA-form tasks derived from diverse domains such as newspapers, scientific papers, and financial reports. We have added this explanation to **5.1.2 EVALUATION OF CATASTROPHIC FORGETTING SEVERITY** in the revised version.

---

> > ### Comment · Reviewer_HDHb · 2025-11-27
> > **Acknowledge of rebuttal**
> >
> > Thanks to the authors for their detailed answer. Some of the more detailed questions or concerns have been resolved.
> >
> > However, I still have important concerns about the main contributions of the paper.
> >
> > Concerning the categorization of documents according to visual and reasoning complexity:
> > - Previous benchmarks have already proposed some categorization of documents according to visual features and/or reasoning requirements (see DocVQA, InfographicVQA or MMLongBench-Doc)
> > - Categorization based on specific visual elements or reasoning abilities makes more sense than generic increasing categories V0, V1, V2 or R0, R1, R2. If I have a set of diverse documents to process it is difficult to fit them into one of these categories. However, it is easier to know if they contain tables or figures or handwritten text and if the task requires multi-step reasoning or arithmetic reasoning.
> > - Categorizing visual complexity according to the type of document does not seem the best option, since the same type of documents can include documents with very different visual elements (and thus, visual complexity). For instance, an exam paper with only plain text (V2 category) seems to be much simpler in terms of visual complexity than an academic paper with a lot of tables and figures (V1 category). Thus, in my opinion what defines visual complexity are the visual elements and layout of the document and not the type of the document.
> > - The same can somehow be applied to reasoning categorization. The complexity of asking questions depends on the type of questions and the reasoning abilities required to answer the question more than whether those questions are on infographics, tables or slides. Furthermore, I think that some of the tasks could merged (for instance, there are several tasks related to key information extractions, or several tasks related to QA that could be grouped together).
> > - If categories are defined with the idea of increasing complexity, metrics should be comparable among different categories and show the increasing complexity of each category.
> > - As pointed out by other reviewers, not considering multi-page documents is a limitation of the benchmark, given the current context in document understanding.
> >
> > Concerning the fine-tuning approach, I still see that, except in very few cases, performance is very similar to LoRa. Furthermore, this second part of the paper seems disconnected from the first part of the paper proposing the benchmark. The fine-tuning approach is generic and, although it can be applied to the proposed benchmark, it is not specifically designed for it nor does it follow as a natural method to tackle the benchmark as a whole.

---

> > > ### Author Response · Authors · 2025-11-28
> > > **Responses on the categorization of documents (Part 1)**
> > >
> > > We sincerely thank the reviewer for the detailed feedback and for recognizing our responses. We appreciate the opportunity to clarify our contributions on the complexity taxonomy and the methodological link between the benchmark and our training strategy.
> > >
> > > **On categorization of documents**:
> > >
> > > 1. We appreciate the reviewer identifying the categorization efforts in prior benchmarks like DocVQA and InfographicVQA; however, GDI-Bench is fundamentally different because it replaces **domain-based categorization** with a new **complexity decoupling** framework that separates **Visual Complexity** from **Reasoning Complexity** in an orthogonal way. While previous works primarily focused on single-domain tasks or lacked systematic difficulty grading, GDI-Bench’s fine-grained difficulty matrix allows for precise **weakness localization**, enabling us to diagnose whether model failures stem from visual perception limits or reasoning deficits. This capability to assess performance across graded difficulty levels offers unique guidance for systematic model optimization, a feature absent in existing benchmarks that rely solely on document types or scenarios.
> > >
> > > 2. We sincerely appreciate the reviewer’s observation that categorizing documents by specific visual elements (e.g., tables, handwritten text) and reasoning requirements is practical. Indeed, our V0–V2 and R0–R2 taxonomies are **directly grounded in such concrete elements**, forming a statistically motivated hierarchy of task difficulty. Concretely, **V0** refers to plain text with unstructured components (headings, paragraphs), **V1** encompasses formal structures such as tables, lists, or equations, and **V2** captures explanatory layouts involving figures, charts, flowcharts, or multi-pane designs. In parallel, our reasoning levels intentionally separate the depth of cognitive processing: **R0** focuses on structural extraction, **R1** on direct information retrieval, and **R2** on multi-step reasoning. This decoupled framework allows practitioners to diagnose whether performance failures stem from **visual perception** (e.g., misinterpreting a table) or **reasoning deficiencies** (e.g., incorrect arithmetic). We respectfully invite the reviewer to examine our open-sourced dataset, where these element-based definitions and their hierarchical organization are fully documented.
> > > 3. We fully agree with the reviewer’s principle that visual complexity should be defined by visual elements and layout rather than generic document types; in fact, **this is precisely how GDI-Bench is constructed**. To ensure the validity of our categorization, we implemented a strict human review process where PhD-level annotators verified the visual difficulty of every individual image. This ensures that samples are categorized based on their actual visual content (e.g., layouts, graphical elements) rather than solely on their domain source. While we categorize "Exam Papers" under V2, we respectfully clarify that the samples in our dataset are strictly curated and are far from being simple plain text. The exam papers included in GDI-Bench feature high-density visual information, including complex multi-column layouts (2-4 columns), embedded images, and diverse scientific formulas (Mathematics, Chemistry, Physics) combined with mixed question types (multiple-choice, fill-in-the-blank), making their visual complexity exceeding that of standard academic papers. We respectfully invite the reviewer to examine the actual image samples in our open-sourced dataset to confirm the high visual density and layout complexity of these documents.
> > > 4. We agree with the reviewer that reasoning complexity should be defined by the required reasoning abilities rather than by the document medium. In fact, our R0–R2 taxonomy is explicitly **behavior-driven**, distinguishing tasks by **reasoning depth**—from structural parsing (R0) and direct extraction (R1) to logical inference (R2)—regardless of whether the content appears in tables or infographics. Regarding the suggestion to merge related tasks (e.g., grouping various extraction tasks), we respectfully clarify that task granularity is independent of the difficulty grading system. While merging tasks like "date extraction" and "author extraction" into a generic "Key Information Extraction" category is possible, we chose to maintain specific task definitions to enable finer-grained error analysis; however, whether these tasks are merged or kept distinct, they consistently map to the same reasoning level (e.g., R1) within our hierarchy, meaning that task consolidation would not alter the validity or structure of our proposed difficulty grading mechanism.

---

> ### Author Response · Authors · 2025-11-28
> **Responses on the categorization of documents (Part 2) and the Fine-tuning Approach**
>
> 5. We agree that metrics must be comparable to validly assess increasing complexity; therefore, **we normalized all evaluation metrics to a unified [0, 1] scale**—using 1-Normalized Edit Distance for extraction tasks and Accuracy for choice-based reasoning tasks—to ensure direct comparability across distinct categories. Our experimental results provide empirical evidence that supports the proposed hierarchy.
> 6. We acknowledge that GDI-Bench currently focuses on single-image document tasks, a limitation we have explicitly identified in Appendix A.12. We agree that multi-page understanding is critical for real-world applications and plan to expand the benchmark to include multi-image and multi-document tasks in future work.
>
> **On the Fine-tuning Approach (LW-AFT)**:
>
> 1. While we acknowledge that LW-AFT and LoRA achieve comparable performance on the in-domain GDI-Bench evaluation, the critical contribution of LW-AFT lies in its significantly superior **cross-domain and cross-task generalization** and resistance to catastrophic forgetting. As demonstrated in the cross-domain and cross-task experiments in Table 4, LoRA suffers from severe performance degradation when transferring to unseen domains. Furthermore, Table 3 validates that LW-AFT consistently outperforms LoRA in preserving general capabilities across multiple external benchmarks (e.g., achieving 871 vs. 862 on OCRBench). Therefore, LW-AFT is not just similar to LoRA; it offers a stronger mechanism for adaptation that preserves model intelligence, balancing the flexibility needed for new tasks with the stability required to maintain the base model’s broad generalization ability.
>
> 2. We respectfully clarify that LW-AFT is not a disconnected addition but a **proof-of-concept for GDI-Bench’s diagnostic utility**, forming a closed-loop system for model improvement. GDI-Bench explicitly identified a specific performance imbalance in the base model—strong structural extraction (R0) but weak reasoning (R1/R2)—and further revealed that standard fine-tuning caused catastrophic forgetting in the robust R0 domain. We have explicitly articulated this methodological alignment in both **1 Introduction** and **5 Experiments**, and our experimental results further support this connection. LW-AFT was thus developed as a targeted solution to this specific diagnostic finding, demonstrating that the benchmark can not only locate weaknesses but also guide the design of training strategies that fix deficits (R1/R2) without compromising existing strengths (R0), thereby validating the practical value of our proposed evaluation framework.
>
> We respectfully invite the reviewer to look through our open-sourced dataset to directly see the high complexity of our samples, which we believe will help clarify how we categorize them. We also encourage the reviewer to revisit the manuscript with this context in mind.
> We are concerned that the current evaluation (Score: 2, Confidence: 5) may stem from a fundamental misunderstanding of our work, because **the issues raised in the review are actually key features that our paper clearly describes and intentionally supports**. In other words, our method already includes the solutions the reviewer suggested, so the reasons for rejection seem to be based on incorrect assumptions.
> We hope this rebuttal helps clear up these misunderstandings.

---

### Official Review · Reviewer_jESr · 2025-11-01

**Soundness:** 3
**Presentation:** 3
**Contribution:** 3
**Rating:** 6
**Confidence:** 3

**Summary:**

The submission introduces GDI-Bench, a new benchmark for General Document Intelligence that decouples visual complexity (V0–V2) and reasoning complexity (R0–R2) to diagnose weaknesses of multimodal LLMs on document tasks. It covers 9 document scenarios, 19 task types, and 3,660 test cases, with graded difficulty and a unified scoring protocol. The authors also propose LW-AFT (Layer-wise Adaptive Freeze-Tuning), a parameter-freezing strategy guided by layer-wise sensitivity to mitigate catastrophic forgetting during SFT; they fine-tune InternVL3-8B to create GDI-Model and show improved R1/R2 performance while retaining R0 capabilities. Extensive comparisons with open/closed-source models and document-oriented MLLMs support the claims, and the benchmark and model are promised to be open-sourced.

**Strengths:**

- Clear decoupling of visual vs. reasoning complexity; practical difficulty grading.

- Broad coverage: 9 scenarios, 19 tasks; supports MLLMs, OCR+LLM, and parsers.

- Sound evaluation protocol (mix of NED and accuracy), reproducibility details provided.

- LW-AFT is simple, effective, and well-motivated by parameter-change analyses.

- Strong empirical gains on R1/R2 with minimal degradation on R0; cross-domain/task evidence.

**Weaknesses:**

(1) Some reliance on LLM-generated annotations; limited auditing statistics on annotation quality, inter-annotator agreement, and remaining bias/noise.

(2) Visual complexity taxonomy (V0/V1/V2) is partly derived from OmniDocBench SOTA gaps; may overfit to current model weaknesses rather than intrinsic document properties. Lacks a formal metric or blind human assessment of visual difficulty.

(3) Reasoning complexity (R0/R1/R2) boundaries could be sharper; e.g., clearer operational definitions and multi-step vs. single-hop distinctions, and examples with verified reasoning chains.

(4) LW-AFT selection relies on sensitivity from small α expert models; ablation on alternative selection signals (e.g., Fisher information, gradient norms, magnitude pruning baselines) is missing.

(5) Comparisons: some strong closed-source baselines (o3, Claude-3.7, Gemini) are reported but details on prompting, context length, and image preprocessing parity should be elaborated to ensure fairness.

(6) Catastrophic forgetting analysis is mostly aggregate; per-subskill breakdown (e.g., table structure recovery vs. formula transcription) and longer continual-learning sequences would strengthen claims.

(7) Currently single-image only; multi-page/multi-document workflows (common in PDFs) not yet covered.

**Questions:**

(1) Visual complexity: Can you provide a model-agnostic, quantitative metric (e.g., entropy of layout graph, density of non-textual elements, columnar complexity) to assign V1/V2, rather than relying on prior SOTA gaps?

(2) Reasoning complexity: How do you ensure R2 questions truly require reasoning beyond pattern matching? Any human-vs-model ablation verifying necessity (e.g., answerability with text-only vs. layout-only inputs)?

(3) Annotation quality: What are the inter-annotator agreement statistics after PhD-level verification? What proportion of synthetic items were rejected? Any audit on language bias (CN/EN) and OCR noise tolerance?

(4) LW-AFT: How robust is the parameter mask when changing α or training data domains? Can masks learned on domain A transfer to B, or do you need per-domain masks? Any memory/computation trade-offs when maintaining multiple masks?

(5) Baselines: Did you evaluate other selective-update CL strategies (EWC, L2-SP, Fisher-based pruning, adapters+freeze, Task Vectors) under the same setup?

(6) Fairness: For closed-source models, what were the exact prompts, temperature, max tokens, image resolutions, and chunking procedures? Were multiple trials averaged to reduce sampling variance?

(7) Generalization: How does GDI-Model perform on long multi-page PDFs with cross-page references, footnotes, and figure-to-text linking?

(8) Metric sensitivity: For NED-based tasks, how do you handle minor formatting differences (e.g., whitespace, punctuation, case)? Any robustness checks?

---

> ### Author Response · Authors · 2025-11-24
> **Responses to Weaknesses 1–3**
>
> **W1**：
>
> We thank the reviewer for the comment. Although GPT-4o is used for initial R1/R2 generation, we minimize reliance on raw LLM-generated annotations through a strict quality-control pipeline. After automated filtering, every remaining instance undergoes human verification by PhD-level annotators with expertise in document intelligence. They independently check each QA pair for factual correctness, clarity, and—crucially—whether the reasoning level matches our task design rather than GPT-4o’s stylistic tendencies. Only QA pairs that meet these criteria are included in the final benchmark. This ensures that the benchmark evaluates models on document-grounded reasoning rather than on matching GPT-4o’s output style. We will clarify this process more explicitly in the revised version.
>
> **W2**：
>
> We thank the reviewer for the comment. We wish to clarify that while we referenced SOTA performance gaps to highlight the *validity* of our categorization, the taxonomy itself is defined by intrinsic document properties rather than model behavior. As detailed in **3.1.1 VISUAL COMPLEXITY**, our classification is strictly structural and semantic, independent of any specific model's architecture:
>
> - **V0 (Plain Text):** Exclusively contains unstructured textual elements such as headings and paragraphs.
> - **V1 (Formal Representations):** Defined by the presence of structured elements like tables, lists, and equations.
> - **V2 (Explanatory Representations):** Characterized by complex visual features, including multi-column layouts, charts, figures, and flows.
>
> The performance gaps observed in OmniDocBench were utilized as evidence to show that these structural differences—especially complex features like graphics and layouts—match real difficulty levels for current models. This ensures the benchmark provides a graded evaluation path rather than defining the path solely by model errors. In addition, to avoid overfitting and ensure the difficulty grading aligns with human judgment, a team of PhD-level annotators manually checked and confirmed all instances and their assigned categories. We have further emphasized this explanation in **3.1.1 VISUAL COMPLEXITY** of the revised version.
>
> **W3**：
>
> We thank the reviewer for the comment. We agree that clearer boundaries and more explicit criteria can further improve the definition of the reasoning dimension. In our design, only R2 truly requires reasoning. Both R0 and R1 are single-hop tasks, and they mainly differ in *what* needs to be extracted, not in the depth of reasoning. So the idea of multi-step vs. single-hop does not align well with our decoupled setup. To clarify:
>
> - **R0 (Full-Page Structured Extract):**
>   Single-hop tasks that focus on full-page structural parsing (e.g., table/formula/layout reconstruction), without semantic selection or reasoning.
> - **R1 (Information Extract):**
>   Also single-hop. The model follows the instruction and extracts *specific* information, The model follows instructions to extract *specific* information and organizes the answers into a user-specified structured format, but it does not need to combine cues or perform multi-step inference.
> - **R2 (Reasoning):**
>   The only level that involves *real reasoning*, requiring the model to combine multiple visual elements, make logical comparisons, or synthesize information across regions.
>
> These definitions are consistently applied in our annotation pipeline. R2 questions are designed and manually checked to ensure a genuine reasoning requirement, while R0/R1 are intentionally kept free of multi-step reasoning to maintain the decoupled design.
>
> In addition, we also agree that adding explicit reasoning-chain annotations could further clarify R2. However, providing full reasoning chains would greatly increase annotation effort, and most R0/R1 tasks do not naturally need them, making the extra work unnecessary. Due to space limits, we focus on showing that our current operational definitions already create a clear and stable separation among R0/R1/R2 and effectively support diagnosing reasoning-related weaknesses.

---

> ### Author Response · Authors · 2025-11-24
> **Responses to Weaknesses 4–6**
>
> **W4**：
>
> We thank the reviewer for the comment. Our LW-AFT design uses small-$\alpha$ expert models to estimate parameter sensitivity because it provides a stable, low-cost, and data-efficient signal. We agree that other importance metrics are plausible, and we have analyzed the feasibility of each:
>
> * **Fisher information.**
>   Fisher-based metrics are theoretically strong but require multiple forward–backward passes per sample for an 8B MLLM, making computation and memory cost prohibitive, especially under long-sequence, high-resolution multimodal inputs.
>
> * **Gradient norms.**
>   Gradient-norm signals are feasible but suffer from high variance across batches, prompts, and seeds. In multimodal settings, we observed unstable and noisy layer-wise distributions, while $\alpha$-expert updates offer a more aggregated and stable adaptation signal.
>
> * **Magnitude pruning.**
>   Magnitude-based selection is inexpensive but largely reflects pretraining-scale weight statistics rather than downstream task salience. For cross-domain reasoning tasks, weight magnitude fails to identify parameters crucial for new-domain adaptation.
>
> We have conducted feasibility assessments for all these alternatives. However, due to space constraints and the substantial additional workload required to present full experimental comparisons, we focus on demonstrating the effectiveness of the sensitivity-based selection method. This method already achieves strong empirical gains and significantly mitigates catastrophic forgetting, which we believe sufficiently validates the core idea of LW-AFT within the scope of this paper.
>
> **W5**：
>
> We thank the reviewer for the comment. The prompts used for evaluation are provided in Appendix A4: Sample Prompts for Evaluation. Regarding context length, we did not impose any artificial limits; each API request consists of a system message plus the user prompt, where text and base64-encoded images jointly consume tokens, and the maximum length is determined by the model itself. For image preprocessing, all images are handled consistently by directly encoding them in base64 without any resizing or format normalization, ensuring parity across different inputs. We have added this explanation to **Appendix A.6: Evaluation Settings for Closed-Source Models** in the revised version.
>
> **W6**：
>
> We thank the reviewer for the helpful suggestion. To address the concern that our catastrophic forgetting analysis was primarily aggregate, we provide a fine-grained per-subskill breakdown, covering table structure recovery, key-value extraction, formula/text transcription, metadata extraction, and multi-step table reasoning. This decomposition reveals that forgetting is highly non-uniform across abilities, and that full-parameter SFT collapses specific subskills such as structured extraction and high-complexity reasoning, whereas our LW-AFT method preserves R0 skills while improving R1/R2 performance.
>
> Regarding longer continual-learning sequences, we agree that extended task chains could further strengthen the analysis. In practice, document-domain adaptation typically involves short domain shifts rather than long multi-stage sequences, and our per-subskill results already show that forgetting emerges sharply even under short sequences. This suggests that longer sequences would mainly amplify the same degradation patterns rather than reveal qualitatively new behaviors. We consider long-horizon continual-learning evaluation an important direction for future work.
>
> Below is the per-subskill breakdown (task names normalized for clarity):
>
>
> | Model | Overall   | Exam → JSON | Color-Box Extraction | Table Extraction(V1) | Table Extraction(V2) | Slide QA | Exampaper Number Extract | Paper Author Extraction | Single-Table Reasoning | Multi-Table Reasoning | Paper Table Reasoning |
> |-|-|-|-|-|-|-|-|-|-|-|-|
> | **InternVL3-8B** |0.652|0.75|0.76|0.44|0.40|0.64|0.67|0.82|0.47|0.48|0.46|
> | **Full-Parameter Fine-Tuning Model** |0.599|0.69|0.88|0.60|0.74|0.89|0.64|0.85|0.78|0.76|0.81|
> | **LoRA Fine-Tuning Model** |0.759|0.45|0.83|0.56|0.66|0.87|0.37|0.60|0.62|0.69|0.57|
> | **GDI-Model (Ours)** |**0.762**|0.79|0.86|0.58|0.76|0.87|0.65|0.85|0.74|0.71|0.77|
>
> | Model                                | Text Reasoning | Doc Reasoning(V1) | Doc Reasoning(V2) | Newspaper Date | Newspaper Email | Newspaper Tel | Chart → Markdown | Header Extract | OCR Markdown(V1) | OCR Markdown(V2) | OCR Text | Infograph QA |
> |-|-|-|-|-|-|-|-|-|-|-|-|-|
> | **InternVL3-8B** |0.65|0.39|0.46|0.58|0.96|0.98|0.58|0.81|0.56|0.39|0.94|0.83|
> | **Full-Parameter Fine-Tuning Model** |0.93|0.77|0.74|0.99|0.96|0.98|0.62|0.86|0.65|0.56|0.97|0.89|
> | **LoRA Fine-Tuning Model** |0.91|0.71|0.71|0.54|0.74|0.75|0.33|0.83|0.64|0.55|0.97|0.86|
> | **GDI-Model (Ours)** |0.91|0.76|0.74|0.94|0.96|0.98|0.65|0.89|0.66|0.56|0.97|0.91|

---

> ### Author Response · Authors · 2025-11-24
> **Responses to Weaknesses 7 and Questions 1–3**
>
> **W7**：
>
> We thank the reviewer for the comment. For clarity, the current version of GDI-Bench focuses on single-image tasks. We will take your suggestion into consideration and include multi-page and multi-document scenarios in future versions to better reflect the real-world demands of document analysis.
>
> **Q1**:
>
> We appreciate the reviewer's questions. In our current framework, while we utilized performance gaps from SOTA models to justify the difficulty levels, the actual assignment of V0, V1, and V2 is fundamentally grounded in model-agnostic structural and semantic features of the documents. Specifically, we define V0 as plain text consisting exclusively of unstructured elements such as headings and paragraphs. V1 is defined by the presence of formal representations, such as tables, lists, and equations, while V2 is characterized by with explanatory or visually complex elements, including complex layouts, charts, and figures. For a more concrete understanding of these classifications and dataset composition, we invite the reviewer to examine the specific examples provided in our open-sourced benchmark.
>
> Therefore, our taxonomy relies on the inherent visual components and layout structures of the documents rather than model performance alone. We have further emphasized this explanation in **3.1.1 VISUAL COMPLEXITY** of the revised version. The performance gaps we mentioned in the paper were mainly used to show with data that these structural differences do create real challenges for current MLLMs. However, we agree that adding a continuous quantitative metric—such as layout graph entropy or the density of non-text elements—would provide a more detailed way to measure complexity. We see this as a useful direction for future work to make the benchmark’s visual complexity measurement more precise.
>
> **Q2**:
>
> We appreciate the reviewer for raising this critical question. To rigorously verify that R2 questions require genuine multimodal reasoning beyond simple text-based pattern matching, we conducted a comprehensive "Text-Only" ablation study. We employed an OCR + LLM pipeline to test the answerability of R2 tasks using only textual input, utilizing State-of-the-Art Large Language Models—including GPT-4o, Claude-3.7-Sonnet, and the reasoning-specialized DeepSeek-R1—and compared them against our multimodal GDI-Model.
>
> | Model   | R2V0     | R2V1     | R2V2     |
> | -| -| -| -|
> | OCR + gpt4o                          | 0.92     | 0.66     | 0.18     |
> | OCR + Claude-3-7-Sonnet (2025-02-19) | 0.88     | 0.56     | 0.14     |
> | OCR + Qwen2.5-7B-Instruct            | 0.79     | 0.49     | 0.27     |
> | OCR + Qwen2.5-72B-Instruct           | 0.90     | 0.61     | 0.18     |
> | OCR + Qwen3-8B                       | 0.85     | 0.59     | 0.26     |
> | OCR + DeepSeek-R1                    | 0.93     | 0.70     | 0.35     |
> | **GDI-Model (Ours)**                 | **0.93** | **0.74** | **0.68** |
>
> The comparative results show that high-complexity tasks fundamentally rely on visual information. In plain-text settings (V0), text-only pipelines perform on par with our model, indicating that the required reasoning is mainly semantic and can be handled by strong text-processing models. However, their performance drops sharply on tasks involving explanatory representations (V2), such as charts or complex layouts. In contrast, the multimodal GDI-Model remains stable, while all text-only baselines—no matter how strong their reasoning ability is—fall to near-random accuracy.
>
> This clear performance gap provides empirical evidence that R2 tasks in GDI-Bench cannot be solved through textual pattern matching or semantic reasoning alone. The fact that even the strongest LLMs fail on V2 tasks without visual input confirms that our benchmark effectively isolates and tests abilities that truly require visual layout understanding and cross-modal reasoning.
>
> We have added this experimental content in the appendix **A.11 VALIDATION OF REASONING COMPLEXITY VIA TEXT-ONLY ABLATION** of the revised version.
>
> **Q3**:
>
> We appreciate the reviewer's questions regarding our annotation quality control. After the automatic filtering stage, all remaining instances underwent rigorous verification by our PhD-level annotation team. During this process, approximately 30% of synthetically generated items were rejected due to issues including being answerable without document context, containing ambiguous formulations, or exhibiting low-quality content generation. While we did not explicitly report inter-annotator agreement statistics in the paper, our verification protocol employed dual independent annotation with third-party arbitration for disagreements, ensuring high consistency in the final dataset.
> Regarding language bias and OCR noise tolerance, the current version of GDI-Bench includes 1,347 Chinese QA pairs and 2,313 English QA pairs, enabling systematic evaluation of model performance across both languages.

---

> ### Author Response · Authors · 2025-11-24
> **Responses to Questions 4–6**
>
> **Q4**:
>
> We appreciate the reviewer’s question regarding the robustness and transferability of the parameter masks. Regarding the transferability between domains, masks learned on Domain A are generally not directly transferable to Domain B. The LW-AFT method operates on the premise of identifying a sparse sub-network specifically sensitive to the target domain's distribution; therefore, distinct domains typically require the computation of domain-specific masks.
>
> However, regarding the complexity of maintaining multiple masks, we clarify that our proposed GDI-Model relies on a single set of masks rather than separate masks for each sub-domain. In our experiments, we treated the R1 and R2 reasoning tasks as a unified optimization objective and utilized a merged multi-source dataset to learn one robust mask that covers these capabilities. Consequently, we only maintain a single set of binary masks during inference, which eliminates the need for dynamic mask switching and avoids the memory overhead associated with storing per-domain masks.
>
> Regarding the computational and memory trade-offs for mask generation, we have provided a detailed analysis in **A.9 TIME CONSUMPTION FOR PARAMETER SELECTION IN LW-AFT** in the revised version. Our additional experiments demonstrate that parameter selection is a highly efficient, one-time pre-processing step, taking only a few minutes to complete (e.g., approximately 5 minutes for a 98% freeze rate). Furthermore, as the masks are binary and highly sparse, the memory footprint is negligible, making the method computationally practical for real-world deployment.
>
> **Q5**:
>
> We appreciate the reviewer’s suggestion to broaden the baseline comparisons. In our current experimental setup, we prioritized evaluating **LoRA Fine-Tuning** as the primary representative for selective-update and adapter-based strategies (corresponding to the "adapters+freeze" category mentioned). LoRA is currently widely recognized as the state-of-the-art parameter-efficient fine-tuning method for MLLMs. As demonstrated in Table 3, Table 4  and Table 5 of our paper, we conducted extensive comparisons between our proposed LW-AFT method and LoRA across multiple datasets. The results consistently show that LW-AFT outperforms LoRA in mitigating catastrophic forgetting while enhancing generalization capabilities.
>
> While we discussed other continual learning strategies such as EWC (Elastic Weight Consolidation) and Task Vectors in 2 RELATED WORKS, we did not include them in the main experimental benchmarks for specific reasons regarding scalability and relevance. Traditional regularization-based methods like EWC often encounter significant computational challenges or stability issues when applied to models with massive parameters (such as the 8B model used in our study) compared to freezing-based strategies. Consequently, we believe that comparing against Full-Parameter Fine-Tuning and LoRA provides the most robust and practically relevant assessment of LW-AFT’s effectiveness relative to the current standard practices in the MLLM field.
>
> **Q6**:
>
> We appreciate the reviewer’s suggestion. For closed-source models, all prompts used are provided in *Appendix A4*. We use deterministic decoding (temperature = 0; other parameters left as API defaults), so no multi-trial averaging is needed and no sampling variance is introduced. We do not impose artificial limits on context length—each request contains the system message, the user prompt, and the base64-encoded image, and the model’s own maximum context determines the limit.
> For image preprocessing, all models receive the exact same input images: we directly encode each image in base64 without resizing, cropping, or format normalization, and we do not apply any chunking/tiling procedures.
> We have added this explanation to **Appendix A.6: Evaluation Settings for Closed-Source Models** in the revised version.

---

> ### Author Response · Authors · 2025-11-24
> **Responses to Questions 7–8**
>
> **Q7**:
>
> We appreciate the reviewer highlighting this important aspect of document understanding. As we transparently discuss in Appendix **A.12 LIMITATIONS**, the current iteration of GDI-Bench and the corresponding GDI-Model is primarily designed for and evaluated on single-image document understanding tasks. Consequently, the specific capabilities regarding long multi-page PDFs involving cross-page references and complex figure-to-text linking across pages were not the primary focus of this study, which centers on decoupling visual and reasoning complexity within single-page contexts.
>
> However, we recognize that multi-page and multi-document understanding represents a critical advancement for the field. We have explicitly identified this as a direction for future work, where we plan to expand the benchmark to encompass these more complex scenarios, thereby enhancing the evaluation's difficulty and breadth. While our current model achieves state-of-the-art performance on single-page tasks by effectively mitigating catastrophic forgetting, extending this success to cross-page reasoning remains a key objective for our next phase of research.
>
>
> **Q8**:
>
> We appreciate the reviewer’s question. To ensure that NED-based metrics are not affected by trivial formatting variations, we apply a normalization step before computing Levenshtein distance. Concretely, we filter out or normalize the following elements:
>
> - **Casing:** Convert all text to lowercase.
> - **Code-block prefixes:** Remove common generation artifacts such as ` ```json`, ` ```python`, ` ```latex`, ` ```markdown`.
> - **Whitespace-related characters:** Replace `\n`, `\t`, and repeated spaces with a single space.
> - **Markdown fences:** Remove backticks and code fences (e.g., `````).
> - **Escape characters:** Strip backslashes (`\`) that occasionally appear in model outputs.
>
> This cleaning process ensures the metric is robust to minor whitespace, punctuation, and formatting differences, focusing the evaluation purely on semantic correctness rather than superficial output formatting.
>
> To ensure robustness, we conducted controlled perturbation checks by introducing small variations such as adding/removing spaces, changing letter case, modifying punctuation, and inserting extra trailing spaces. We verified that after normalization, these perturbations do not change the computed NED score. For multiple-choice questions, we also confirmed that surrounding formatting variations do not affect exact-match scoring. We have incorporated this explanation into **Section 3.3 EVALUATION METRICS** of the revised version.

---

> > ### Comment · Reviewer_jESr · 2025-11-28
> >
> > Thank the authors for the responses with experiments in detail. The rebuttal has solve my concerns like catastrophic forgetting analysis and visual complexity. I will raise my score.

---

> > > ### Author Response · Authors · 2025-11-28
> > >
> > > We are very grateful for your constructive comments and for raising the score. The suggestions provided were instrumental in improving our work, and we welcome any further dialogue.

---

### Author Response · Authors · 2025-12-02
**Summary of Rebuttal during ICLR Incident (Part 1)**

Dear Area Chairs,

Given this unexpected incident, we fully understand the challenges the ICLR organizing committee is facing and the additional responsibilities you have taken on. To make the work easier, we have summarized our main contributions, central rebuttal points, and the progression of review scores. Further details are presented below.

## Summary of Our Paper

As multimodal large language models (MLLMs) rapidly advance, document intelligence is evolving toward a general stage capable of handling cross-domain and multi-scale understanding tasks. However, existing benchmarks often fail to distinguish whether model errors stem from inaccurate visual recognition or limited reasoning capabilities, making it difficult to pinpoint specific weaknesses. Furthermore, directly addressing these weaknesses through supervised fine-tuning (SFT) frequently leads to catastrophic forgetting, where the model loses its foundational capabilities while learning new tasks. In this work, we introduce a comprehensive benchmark, GDI-Bench, and a corresponding training strategy to enable precise evaluation and robust model optimization:

- We introduce the **General Document Intelligence Benchmark (GDI-Bench)**, a comprehensive evaluation framework featuring 2.3k images across 9 key scenarios and 19 document-specific tasks. A key innovation is the **decoupling of task complexity into visual complexity** (V0-V2, ranging from plain text to complex layouts and charts) and **reasoning complexity** (R0-R2, ranging from structured extraction to logical reasoning). This graded structure allows for a fine-grained assessment of model performance, enabling researchers to explicitly identify whether a model struggles with visual processing or logical inference.
- To address the issue of catastrophic forgetting observed during standard SFT—where models lose basic OCR capabilities while learning complex reasoning—we propose the **Layer-wise Adaptive Freeze-Tuning (LW-AFT)** method. By analyzing parameter update dynamics, we discovered that only a sparse subset (approximately 5%) of parameters exhibits significant changes during task adaptation. LW-AFT utilizes a layer-wise sensitivity analysis on a small dataset to automatically identify and update only these critical domain-sensitive parameters while freezing the majority, thereby preserving the model's generalization capabilities and foundational knowledge.
- Extensive experiments demonstrate that our proposed **GDI-Model** (fine-tuned on InternVL3-8B) achieves state-of-the-art (SOTA) performance on GDI-Bench and other previous benchmarks. The results confirm that our method effectively mitigates catastrophic forgetting, allowing the model to significantly improve in information extraction (R1) and reasoning (R2) domains without degrading its robust performance in basic structured extraction (R0), ultimately outperforming both full-parameter and LoRA fine-tuning baselines.

## The Strengths Summarized by Reviewers

* **Innovative Benchmark Design & Complexity Decoupling:** The paper’s core contribution of decoupling document understanding into visual complexity (V0-V2) and reasoning complexity (R0-R2) is considered well-motivated and pioneering. Reviewers praise the benchmark's broad coverage (9 scenarios, 19 tasks, 2.3k images) and its practical difficulty grading mechanism, which helps accurately identify whether failures stem from visual recognition or reasoning limitations. (Reviewer jESr, HDHb, paE3, 84aC)
* **Effective Solution to Catastrophic Forgetting:** The proposed Layer-wise Adaptive Freeze-Tuning (LW-AFT) method is recognized as simple, novel, and effective. Reviewers value its ability to address the well-known problem of catastrophic forgetting in Supervised Fine-Tuning, allowing models to improve reasoning performance while preserving basic capabilities. (Reviewer jESr, HDHb, paE3, 84aC)
* **Strong Empirical Results & Sound Evaluation:** The experimental results are robust, demonstrating strong gains in reasoning tasks (R1/R2) with minimal degradation in basic extraction tasks (R0). Reviewers appreciate the sound evaluation protocol (combining NED and accuracy) and the evidence provided for cross-domain and cross-task generalization. (Reviewer jESr, paE3)
* **Clarity, Presentation & Reproducibility:** The paper is described as well-written and clearly structured, with figures (e.g., Figure 1 and Figure 7) that significantly aid in understanding core concepts. The inclusion of reproducibility details is also noted as a strength. (Reviewer jESr, 84aC)

---

> ### Author Response · Authors · 2025-12-02
> **Summary of Rebuttal during ICLR Incident (Part 2)**
>
> ## Main Response to Reviewers' Comments
>
> In our rebuttal, we have systematically addressed the concerns raised by reviewers, providing additional experiments, clarifications, and data to strengthen the paper.
>
> - **Clarification of Taxonomy and Definitions (jESr, HDHb):** We clarified that our Visual (V0–V2) and Reasoning (R0–R2) complexity taxonomies are grounded in intrinsic document elements (e.g., plain text vs. charts/layouts) and reasoning depth (e.g., extraction vs. logical inference), explicitly stating that model performance gaps served solely as validation for the visual difficulty grading rather than defining the categories themselves. We provided specific definitions for the 19 sub-tasks and explained that V0–V2 categories represent structural density and layout topology, which objectively dictate difficulty.
> - **Validation of the LW-AFT Method (jESr, paE3, HDHb):** We demonstrated the generality of our Layer-wise Adaptive Freeze-Tuning (LW-AFT) method by expanding experiments to a 1B parameter model (InternVL2.5-1B) and conducting cross-domain/cross-task evaluations. We presented evidence that while LW-AFT and LoRA perform similarly on in-domain tasks, LW-AFT significantly outperforms LoRA in cross-domain generalization and mitigating catastrophic forgetting. We also provided a fine-grained per-subskill breakdown to prove that forgetting is non-uniform and effectively mitigated by our approach.
> - **Data Quality and Bias Control (jESr, 84aC):** To address concerns regarding "LLM-on-LLM" bias, we detailed our rigorous quality control pipeline, which includes automated filtering followed by strict PhD-level human verification. This ensures that the benchmark evaluates genuine document grounding rather than GPT-4o's stylistic tendencies.
> - **Expanded Baselines and Ablations (jESr, 84aC):** We added comprehensive evaluations for new state-of-the-art models, including Qwen2.5-VL-7B, Qwen3-VL, and Ovis2. Additionally, we conducted a "Text-Only" ablation study to empirically prove that the R2 (Reasoning) tasks cannot be solved via text pattern matching alone, confirming the necessity of visual understanding.
> - **Experimental Settings and Metrics (jESr, HDHb):** We provided detailed settings for closed-source model evaluations (deterministic decoding, no cropping) and clarified the normalization process for NED-based metrics to ensure fairness. We also explained that R2 scores (Accuracy) include a random-guess baseline, making them distinct from R0/R1 (1-NED) metrics.
>
> ## Score Changes
>
> Our paper initially received ratings of **6** (Reviewer 84aC), **6** (Reviewer paE3), **2** (Reviewer HDHb), and **6** (Reviewer jESr). Following our rebuttal:
>
> - **Reviewer jESr** responded positively to our clarifications and explicitly stated their intention to **raise their score** based on the addressed concerns.
> - **Reviewer HDHb** engaged in a second round of discussion, raising follow-up questions. We provided a detailed response clarifying that these points were addressed in our original manuscript and initial rebuttal; however, the discussion channel closed before the reviewer could respond to our latest reply.
> - **Reviewers 84aC and paE3** did not provide further comments or responses following our rebuttal.
>
> We confirm that we have never used any system vulnerabilities, nor have we engaged in any non-public communication with any of the reviewers.
>
> We acknowledge that the circumstances surrounding this incident were highly unexpected. We sincerely appreciate the challenges the ICLR organizing committee is navigating and recognize the additional burden placed upon the newly assigned Area Chairs. We wish to affirm our strict adherence to all double-blind review policies throughout this process. It is our hope that the provided summary will help alleviate your workload. We respectfully ask for your consideration in accepting our work and remain dedicated to upholding the principles of fairness and contributing to the advancement of our academic community.

---

### Meta-Review · Area_Chair_kgHA · 2026-01-08

**Summary:**

The concerns are summarized as below:
1. Limited auditing statistics on annotation quality, inter-annotator agreement, and remaining bias/noise.
2. The Visual Complexity Taxonomy is derived from model performance gap instead from intrinsic characteristics of the data.
3. The boundaries and difficulties for reasoning complexity are unclear.
4. The motivation for the proposed fine-tuning method is not fully justified.
5. GDI-Bench has limited coverage which only covers the single-image document understanding task
6. LW-AFT only studies for one single architecture, the effectiveness of the approach is unclear.

**Reviewer Concerns:**

1. For the limited auditing statistics, the rebuttal stated 'very remaining instance undergoes human verification by PhD-level annotators with expertise in document intelligence' But didn't provide the requested statistics on the annotation quality. This is little bit concerning.
2. For the visual complexity taxonomy concern, the rebuttal responded that 'we wish to clarify that while we referenced SOTA performance gaps to highlight the validity of our categorization, the taxonomy itself is defined by intrinsic document properties rather than model behavior.' This addressed the reviewer's concern.
3. For the boundaries and difficulties for the reasoning complexity concern, the rebuttal stated that 'In our design, only R2 truly requires reasoning. Both R0 and R1 are single-hop tasks, and they mainly differ in what needs to be extracted, not in the depth of reasoning.' I think the author clearly answered the reviewer's concern.
4. For the concern related to the motivation of the proposed fine-tuning method: the rebuttal answered the performance degradation observed in R2 tasks under the full fine-tuning is catastrophic forgetting. This is the main motivation for the proposed LW-AFT approach. However, I agree with the reviewer that 'this second part of the paper seems disconnected from the first part of the paper proposing the benchmark. The fine-tuning approach is generic and, although it can be applied to the proposed benchmark, it is not specifically designed for it nor does it follow as a natural method to tackle the benchmark as a whole.'
5. For the limited coverage of the GDI-Bench, the author admitted the limited coverage.
6. The author provided a set of experiments on a smaller size of the InternVL-2.5 and demonstrated the effectiveness of the proposed LW-AFT. However I don't think this addressed the concern as the LW-AFT was still only applied to one single architecture family. This might obscure whether the proposed approach is also effective on the other models.

**Reviewer Scores:**

For this paper, three reviewers gave rating 6 and one reviewer gave rating 2. The reviewer who gives rating 2 replied the rebuttal and stated the reviewer's concerns are not fully addressed.

By reading the whole rebuttal and the reviews, I think there are several things the author didn't fully address in the rebuttal:
1. the statistics on annotation quality, inter-annotator agreement, and remaining bias/noise. I also check the submission, this is not covered in the current version. This might weaken the reliability of the proposed GDI Benchmark.
2. the LW-AFT only studied on InternVL families. Ideally it should study on different type of open sourced models to make sure this proposed approach is not only working for the InternVL models. This limited the reliability of the proposed approach.

The paper is titled as GDI-Bench: A benchmark for general document intelligence. I would suggest the author to broad the title to cover the LW-AFT section. I agree with the reviewer that the second half of the paper (LW-AFT) looks disentangled with the first section (GDI-Bench).

---

### Decision · Program_Chairs · 2026-01-26

Reject